# First assessment of Iranian pomegranate germplasm using targeted metabolites and morphological traits to develop the core collection and modeling of the current and future spatial distribution under climate change conditions

Maryam Farsi[1], Mansoor Kalantar[1]\*, Mehrshad Zeinalabedini[2]\*, Mohammad Reza Vazifeshenas[3]

1 Department of Plant Breeding, Yazd Branch, Islamic Azad University, Yazd, Iran, 2 Department of Systems Biology, Agricultural Biotechnology Research Institute of Iran (ABRII), Agricultural Research, Education and Extension Organization (AREEO), Karaj, Iran, 3 Yazd Agricultural and Natural Resources Research and Education Center, Agricultural Research, Education and Extension Organization (AREEO), Yazd, Iran

\* mzeinolabedini@abrii.ac.ir (MZ); mkalantar@iauyazd.ac.ir (MK)

## Abstract

Pomegranate has been considered a medicinal plant due to its rich nutrients and bioactive compounds. Since environmental conditions affect the amount and composition of metabolites, selecting suitable locations for cultivation would be vital to achieve optimal production. In this study, data on the diversity of targeted metabolites and morphological traits of 152 Iranian pomegranate genotypes were collected and combined in order to establish the first core collection. The multivariate analyses were conducted including principal component analysis (PCA), and cluster analysis. In addition, the current and future geographical distribution of pomegranate in Iran was predicted to identify suitable locations using the MaxEnt model. The results showed high diversity in the studied morphological and metabolic traits. The PCA results indicated that FFS, NFT, JA, and AA are the most important traits in discriminating the studied genotypes. A constructed core collection using maximization strategy consisted of 20 genotypes and accounted for 13.16% of the entire collection. Shannon-Weaver diversity index of a core collection was similar or greater than the entire collection. Evaluation of the core collection using four parameters of MD, VD, CR, and VR also indicated the maintenance of the genetic diversity of the original set. According to the MaxEnt model, altitude, average temperature of coldest quarter, and isothertmality were the key factors for the distribution of pomegranate. The most suitable areas for pomegranate cultivation were also determined which were located in arid and semi-arid regions of Iran. The geographic distribution of pomegranate in the future showed that the main provinces of pomegranate cultivation would be less affected by climatic conditions by the middle of the century. The results of this study provide valuable information for selection of elite genotypes to develop the breeding programs to obtain the cultivars with the highest levels of

**Data Availability Statement:** All relevant data are within the manuscript and its Supporting information files.

**Funding:** The authors received no specific funding for this work.

**Competing interests:** The authors have declared that no competing interests exist.

metabolic compounds for pharmaceutical purposes, as well as identification of the most suitable agro-ecological zones for orchard establishment.

## Introduction

Pomegranate (*Punica granatum* L.) is one of the oldest known edible fruit tree species, belongs to the Punicaceae family [1], and is native to Central Asia. Iran is known as the center of diversity and likely as primary origin of pomegranate [2], however as the plant is adapted to a wide range of climatic conditions, it is widespread around the world with a great genetic diversity [2]. Iran is one of the largest producers of pomegranates in the world [3] with the significant genotype and cultivar diversity. As one of the most important centers of genetic resources of pomegranate, the collection of Yazd Pomegranate Research Station ranks third in the world with more than 760 Genotypes [4].

Regarding the high nutritional [5], medicinal and economic [4] value of pomegranate, there is a growing global demand for this product. All parts of the tree (fruits, leaves, flowers, and roots) are applied for medicinal purposes [6]. The edible portion of pomegranate fruit (aril) contains significant amounts of sugars, vitamins, polysaccharides, polyphenols, and minerals [7, 8]. Numerous scientific studies indicated that pomegranate contains antioxidant, anti-carcinogenic, anti-inflammatory, anti-diabetic, and antimicrobial compounds [9–13].

While pomegranate genotypes grow in different climate conditions around the world, approximately 10% of genotypes is cultivated commercially [14], however this rate is much lower (about 1.5%) in Iran despite the very high diversity of pomegranates [15]. Accurate identification of genotypes [16, 17] and metabolic profiles [18, 19] were considered as essential components to establish the plant breeding programs. The study of fruit morphological traits is useful in identifying different genotypes of pomegranate [20, 21]. Morphological characteristics of the fruit such as weight and shape of the fruit, the aril juiciness, as well as the number of fruits and yield per tree are important criteria for selecting a genotype with a marketable product and high yield [22]. In addition, information about the metabolic content of different pomegranate genotypes could help us in selecting and producing higher quality cultivars for medicinal purposes. Because metabolites are the result of gene interaction from different ways, metabolomics data can be used to measure genetic diversity, and in combination with molecular markers and morphological trait data facilitate the identification of valuable genetic resources [23]. While molecular markers [21, 24–28] and the morphological traits [29–33] have been studied in a large number of Iranian pomegranate genotypes, the diversity of metabolic compounds has not [34–37].

The establishment of a core collection can minimize the number of genotypes and reduce the preservation costs while representing the entire genetic diversity [38, 39]. Selection of superior genotypes and generating the new cultivars depends on the purpose of fresh consumption or medicinal use will be more feasible in a small germplasm collection.

Climate change is now an important challenge facing agriculture worldwide. The increase of average temperature, changing the amount and pattern of rainfall has altered the climatic classification of different regions of Iran [40], which can affect the suitability of habitats for pomegranate cultivation. Climate change affects the growth and development of plants and, consequently, the quality and quantity of products [41, 42]. The amount and composition of secondary metabolites produced in plants are regulated by environmental factors and, as a result, their production is threatened by climate change [43]. For example, some metabolites accumulate at lower temperatures [44, 45], while some other compounds are produced at

higher temperatures [46]. Species distribution models (SDMs) are suitable tools for identifying the appropriate climatic range for species growth at present and examining the displacement of spatial distribution and unsuitable areas for species growth in climate change conditions. One of the common methods used to study the distribution of plant species based on presence-only data is the MaxEnt entropy method [47–49]. This model reveals the best forecasting capacity and is the most accurate method [50, 51]. Gunawan et al. [52] modeled the geographical distribution potential of the *Baccaurea macrocarpa* fruit tree in southern Kalimantan, Indonesia, and found the MaxEnt model had the better predictive performance.

The objective of this study is to develop a core subset of pomegranate using the targeted metabolite contents and morphological traits in 152 genotypes for the selection of genotypes with the high economic and medicinal value to apply in future breeding programs as well as to predict the current and future potential geographical distribution of pomegranate in Iran which assist in the preparation of a map for orchards establishment in the most suitable agro-ecological zones.

## Materials and methods

One hundred fifty-two accessions were selected out of 760 collected pomegranate genotypes from Yazd pomegranate collection in Agriculture and Natural Resources Research Centre of Yazd (31° 55′ N, 54° 16′ E, and 1216 m alt.) (S1 Table). The results of Kazemi alamuti et al. [26], Mousavi Derazmahalleh et al. [28], Razi et al. [33] studies were used to select the genotypes. This site has an average annual minimum and maximum temperature of 4 and 20.5°C, respectively, with an annual rainfall of 96 mm.

### Investigation of morphological characteristics

Twelve morphological traits of pomegranates were used to study the diversity of genotypes, which included the number of fruit per tree (NFT), mean fruit weight (FMW), mean tree yield (TMY), anthocyanin on the branch of this year (ABTY), petiole color (PC), fruitful flower size (FFS), Fruit albedo color (FAC), fruit bottom shape (FBS), fruit heel shape (FHS), juice content of arils (JA), seed size (SS), and seed color (SC). The number of plants evaluated for each genotype was three trees from which eight samples for each genotype were randomly harvested, and transferred to the laboratory for assessment. Fruit sampling was performed during their ripening stage. Qualitative traits were evaluated visually and then scored (Table 1). As the

**Table 1. Scoring morphological traits of pomegranate.**

| trait | Abbrev. | Scoring |
|---|---|---|
| Anthocyanin in branch of this year | ABTY | = 1 = None, 3 = little, 5 = medium, 7 = much |
| Petiole color | PC | 1 = Red, 2 = red-orange, 3 = light red, 4 = dark red |
| Fruitful flower size | FFS | 3 = small, 5 = medium, 7 = large |
| Fruit Albedo Color | FAC | 1 = white, 2 = Cream, 3 = yellow, 4 = pink, 5 = Purple |
| Fruit Heel shape | FHS | 1 = Without heel, 2 = short heel, 3 = high heel |
| Fruit Bottom shape | FBS | 1 = round, 2 = Convex, 3 = Angled, 4 = plate |
| Juice content of arils | JA | 1 = low juicy, 3 = normal, 5 = juicy |
| Seed Color | SC | 2 = white to pink, 3 = pink, 4 = light red, 5 = red, 6 = dark red |
| Seed size | SS | 3 = small, 5 = medium, 7 = large |
| Number of fruits per tree | NFT | 1 = 0–9, 3 = 10–19, 5 = 20–29, 7 = 30–39, 9 = 40–49 |
| Mean fruit weight | FMW | 1 = 0–99, 3 = 100–199, 5 = 200–299, 7 = 300–399, 9 = 400–499 |
| Mean tree yield | TMY | 1 = low, 3 = medium, 5 = high |

preliminary analysis of data for two consecutive growing seasons (2017–2019) showed no significant difference, the average of two years of data was used for statistical analysis to increase the accuracy in the analysis.

## Biochemical properties evaluation

The pH measurement of pomegranate aril juice was performed using a digital pH meter (Metrohm model 601) at 21˚C.

Total soluble solids in pomegranate extracts were measured using a Three-in-one Digital Refractometer (model MTD045Nd) with a temperature correction of 0 to 45% Brix amplitude at 20˚C and indicated in Brix.

Acidity was measured in terms of citric acid using the G-Won acidometer (model GMK-825) by the method defined by Horwitz and Latimer [53].

The concentration of total anthocyanin was calculated by a pH-differential method using two buffer systems described by Tzulker et al. [54]. Total anthocyanin content was calculated using the following formula [55]:

$$\text{Anthocyanin pigment (mg/L)} = \frac{A \times MW \times DF \times V \times 1000}{a \times l \times m}$$

where A is the absorbance, MW is the molecular weight of cyanidin-3-glucoside (449.2 g/mol), DF is the dilution factor, V is the solvent volume (mL), $a$ is the molar absorptivity (26,900 L. $mol^{-1}.cm^{-1}$), and $l$ is the cell path length (1 cm), and $m$ is the freeze-dried sample weight (g).

The total polyphenol content was determined by the Folin-Ciocalteu method [56]. In this method, 100 ml of diluted pomegranate juice (25:100) with methanol: water (6: 4) was mixed with 100 ml of folin and 1.58 ml of distilled water. Then 300 μL of 7.5% sodium carbonate solution was added and its absorbance was measured at 760 nm after 90 minutes at room temperature. In this method, gallic acid was used as standard.

Antioxidant activity was evaluated by Brand-Williams et al. [57] method. In this method, 2 ml of 0.1 mM DPPH solution was added to 100 μl of diluted pomegranate (1:100) with methanol: water (6:4), and the absorbance was maintained for 30 min at room temperature. It was read at 517 nm using a UV-Visible spectrophotometer (Cary 300 Scan) and relative absorbance of the control sample (0.1 mM DPPH pomegranate-free solution). The antioxidant activity was calculated according to the following formula [58]:

$$\textit{Antioxidant activity } (\%) = [1 - (\textit{Abs sample } 517 \textit{ nm } / \textit{ Abs control } 517 \textit{ nm})] \times 100$$

## Development of primary core collection

In this study, morphological traits combined with metabolic data were considered for the core collection development. Data were analyzed using maximization strategy and modified heuristic algorithm to create a core collection [59] in PowerCore 1.0 software. This software selects the genotypes based on the maximum diversity through a modified heuristic algorithm, which represents the complete coverage of the traits in the entire set.

To evaluate the core collection, four indicators were calculated, namely, the mean difference percentage (MD), the variance difference percentage (VD), the coincidence rate of range (CR), and variable rate (VR) of coefficient of variation for testing the diversity and representativeness of the primary core collection. A core collection with MD less than 20% and CR more than 80% was considered as a representative set. In addition, higher values in VD and VR were considered to represent a more efficient core set [60]. Shannon-Weaver diversity index between the total population and the core collection was also calculated for each trait.

## Statistical analysis

Descriptive statistics were carried out for selected metabolites and morphological traits of the genotypes. Coefficients of variation and the Shannon–Weaver (H') index were calculated to evaluate the diversity of morphological traits measured among genotypes.

In order to determine correlations between morphological and metabolic characteristics, a Spearman correlation was performed using "corrplot" package in software R.

Morphologic and metabolic data were analyzed using cluster analysis based on Euclidean distance and Ward's method and displayed as heatmap too. In addition, the population structure of 152 pomegranate genotypes used in this study was evaluated through a model of Bayesian clustering algorithm using the "apcluster" package in software R. The optimum number of subpopulations (K) was determined by Calinski-Harabasz criterion (CHC) to the K-means clustering.

Principal Component Analysis (PCA) was employed on all variables simultaneously to identify traits that contributed the most variability within a group of genotypes. All analyses were conducted by using R-software.

## Potential geographical distribution prediction of pomegranate under current and future conditions in Iran

The coordinates of the 152 points representing the spatial distribution of pomegranate genotypes originating from different provinces in Iran were determined and denoted on the map (Fig 1). Seventy-five percent of location data was randomly selected to create the training model, and the remaining 25% of data was used for model validation.

The 19 bioclimatic variables and altitude were used to model the potential distribution areas under the present (1970 to 2000) and future (2050) climate conditions. The current

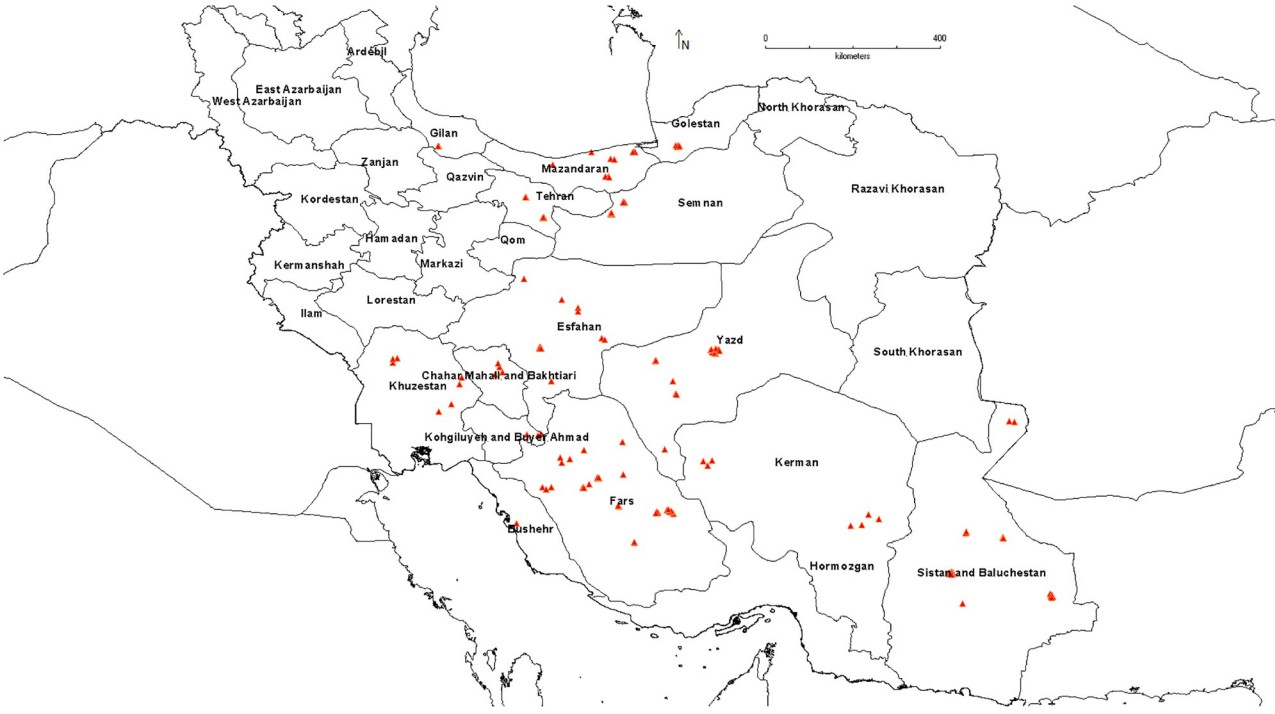

**Fig 1. Locations of 152 distribution points of the studied pomegranate genotypes in Iran.**

climatic data were collected from the WorldClim dataset (http://www.worldclim.org/) with a spatial resolution of 30 s (ca. 1 km$^2$). The data was converted into an ESRI ASCII GIS (.asc) file by DIVA-GIS for the software. Future potential distribution areas were identified using climate layers based on the projections of the Community Climate Model (CCM ver. 3) over the period 2050. The maximum entropy model (MaxEnt v3.3.3) was utilized to predict the pomegranate distribution in Iran.

The relative importance of the selected climate variables on the distribution of pomegranate was evaluated by Jackknife test in which a higher gain value indicates a better fit of the environmental factor in the model [61, 62]. To evaluate the accuracy of the model, the receiver operating characteristic curve (ROC) was plotted by MaxEnt. An area under the curve (AUC) was calculated. Model performance was categorized as failing (0.5–0.6), poor (0.6–0.7), fair (0.7–0.8), good (0.8–0.9), or excellent (0.9–1.0) [63].

## Results and discussion

### Investigation of morphological traits diversity of genotypes

Since the characteristics related to yield, fruit shape, and the juiciness of aril are important for the commercial production of pomegranate, these traits were evaluated in the genotypes (S1 Table). Analysis of morphological traits showed that there was a remarkable variation for the fruit heel and lower fruit shape, seed color, petiole color, mean fruit weight, and the number of fruit per tree (with coefficients of variation of 49.8, 49.3, 44.5, 44.0, 32.0, and 30.6, respectively) in the germplasm of Iranian pomegranate. The coefficient of variation (CV) is a beneficial statistic for comparing the diversity of a morphological trait among genotypes.

According to Shannon index, the fruit bottom shape (1.3), seed color (1.55), mean tree yield (1.06) and the number of fruits per tree (0.92) had the highest diversity in the studied traits.

Fruit size is one of the most important characteristics and the most variable morphological traits among genotypes. The highest value of average fruit weight belonged to three genotypes of 123, 108, and 110 from Sistan and Baluchestan province. The highest number of fruits per tree and average tree yield were obtained from some genotypes of Sistan and Baluchestan, and Fars provinces. The geographical distribution map of genotypes in terms of CV of yield also showed that the highest variability of yield was observed in genotypes of Fars, Sistan and Baluchestan, and Isfahan provinces (Fig 2).

Research on morphological characteristics of pomegranate showed a very large variability among pomegranate genotypes for the studied traits [17, 30, 33, 64]. Tapia-Campos et al. [65] studied on 21 fruit traits of 18 pomegranate genotypes in southern Jalisco, Mexico. They found that fruit size and weight were the most important variables. Morphological traits of 117 pomegranate genotypes in Yazd province, Iran showed that the fruit bottom shape and the fruit shape had high diversity based on the Shannon index [30]. Karapetsi et al. [66] also reported that fruit weight had a high variability (CV = 30.5%) among the studied pomegranate genotypes. In the present study, the attribute of JA was also affected by genotype, which corresponds with the results of Tehranifar et al. [35] and Hmid et al. [67] who also reported the aril juice percentage varied between cultivars.

### Biochemical properties evaluation

It is important to study the metabolic content of fruits of different pomegranate genotypes for use in pharmaceuticals. In addition, the key indicators for evaluating pomegranate flavor are acidity and total soluble solids, which assist breeders to select superior genotypes for fresh eating and the fruit juice industry. The results indicated that the metabolite content of

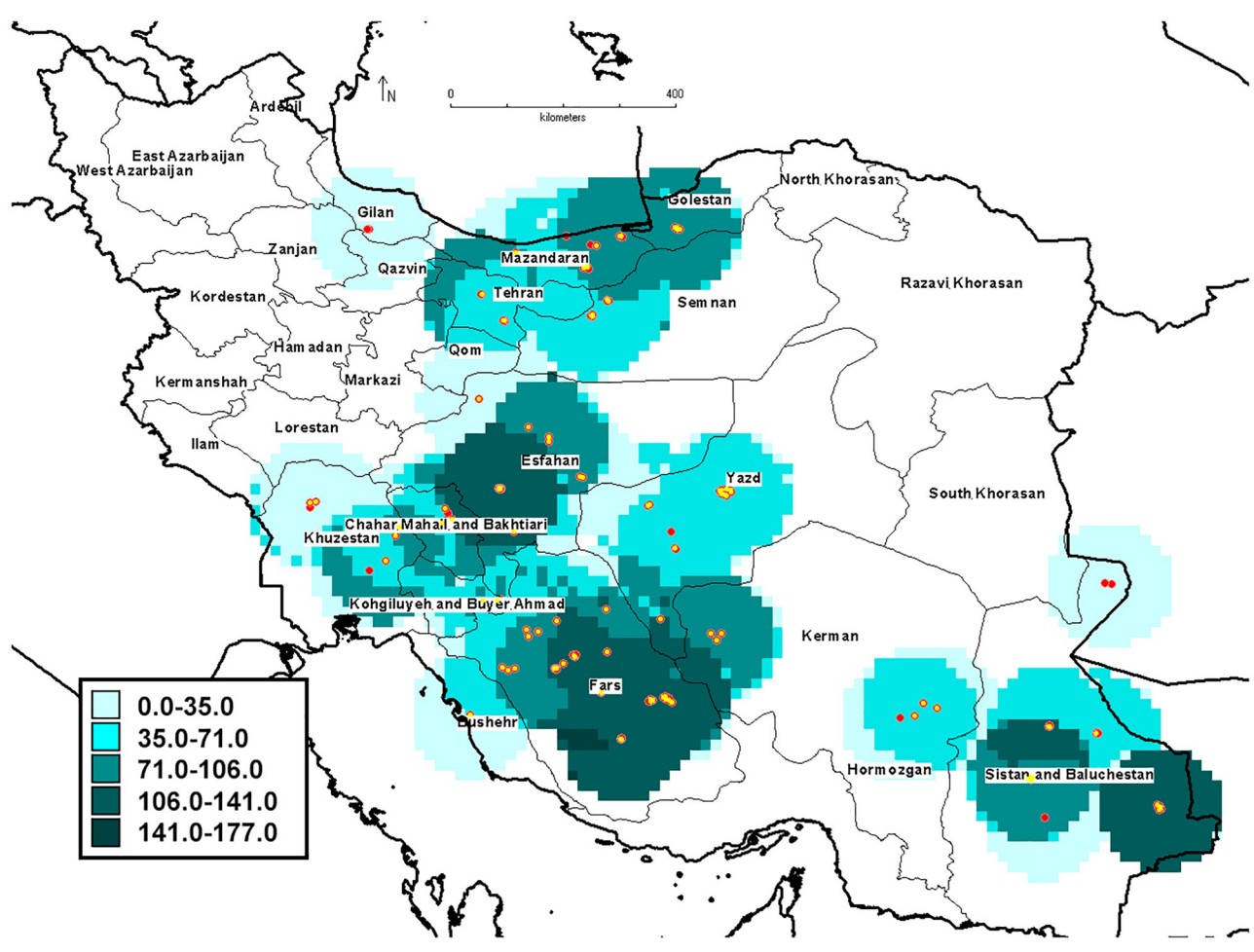

**Fig 2. The geographical distribution map of pomegranate genotypes evaluated in terms of yield (CV).**

pomegranate juice was affected by genotype as values of the studied metabolites content showed considerable variations among genotypes both within and between provinces (S1 Table, Table 2). The highest metabolic content was obtained in Kerman and Fars genotypes. The metabolic content also showed that Yazd genotype had the highest variation of metabolic content too (Fig 3). Kazemi alamuti et al. [26] in the study of genetic diversity of 738 Iranian pomegranate genotypes also stated that the highest variation was also observed in the genotypes of Yazd province. The genotype, climatic conditions, and physiological stage of fruit

**Table 2. Descriptive statistics of the biochemical traits of 152 pomegranate genotypes.**

| Trait | Unit | Minimum | Maximum | Mean | CV. |
|---|---|---|---|---|---|
| Total anthocyanin | mg g$^{-1}$ | 0.17 | 55.13 | 25.50 | 68.66 |
| Antioxidant activity | % | 5.23 | 97.82 | 39.15 | 43.36 |
| Total phenol | mg L$^{-1}$ | 269.5 | 7331.66 | 1676.59 | 56.83 |
| Acidity | g L$^{-1}$ | 0.46 | 2.20 | 0.92 | 34.44 |
| Total soluble solids | % | 8.8 | 19.27 | 15.19 | 12.45 |
| pH | - | 2.14 | 4.02 | 2.94 | 13.94 |

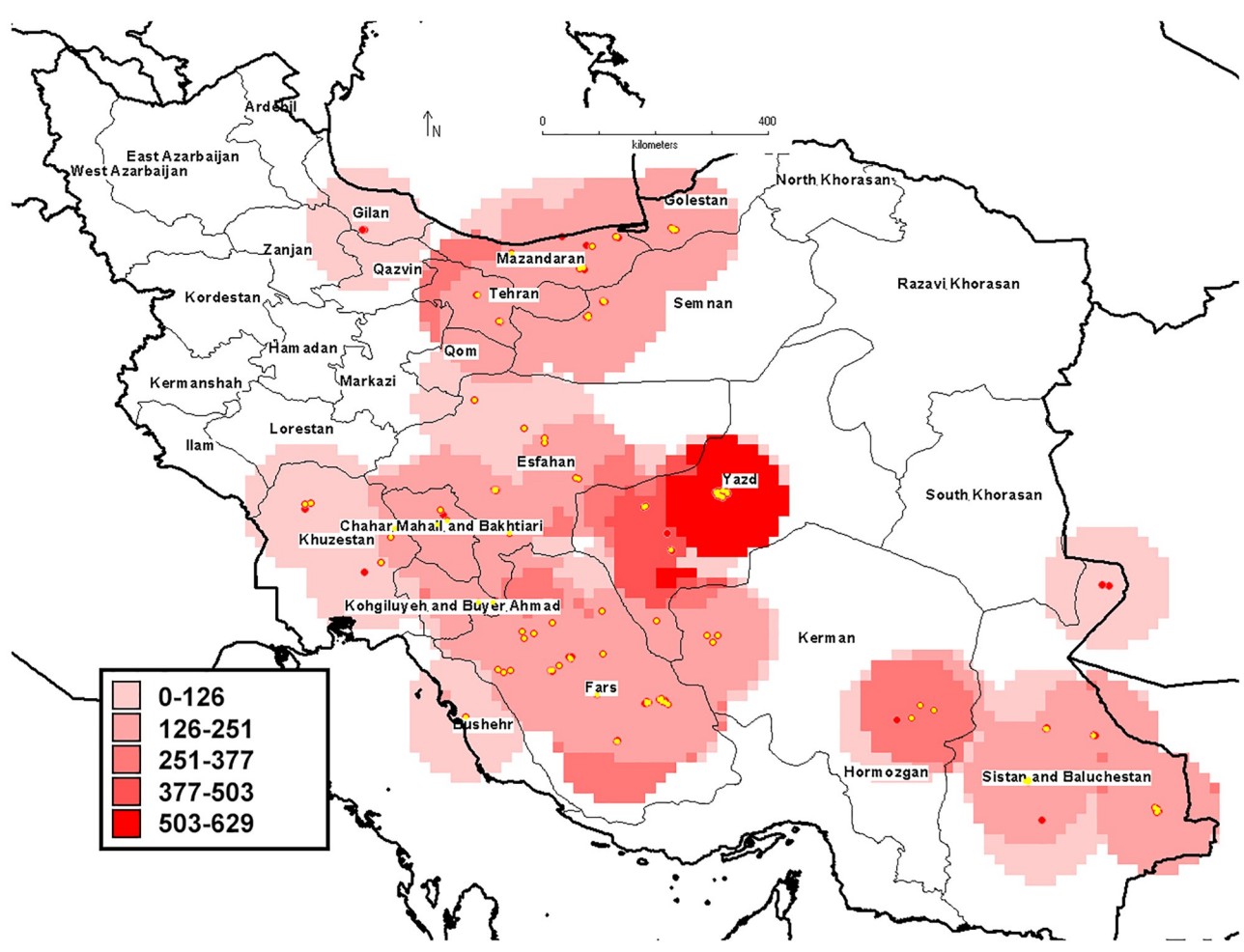

**Fig 3. Distribution map of 152 pomegranate genotypes based on biochemical properties (CV).**

growth and fruit placement on the tree had the significant effect on the concentrations of pomegranate fruit metabolites [68, 69]. In addition, variation of biochemical properties of genotypes of different regions indicated that environmental conditions might also influence the biochemical compound content in pomegranate which was observed in several studies [29, 64, 70].

The evaluation of anthocyanin content among the genotypes of pomegranate showed a relatively high level of variability (range of 0.17 to 55.13 mg g$^{-1}$). In this research, a large number of genotypes including commercial pomegranate, wild genotypes and completely colorless aril genotypes were studied. As anthocyanins are responsible for the color of many fruits [71], including pomegranates, the anthocyanin levels of aril juice were very low in some of the studied genotypes. Besides, fruit storage conditions before and during the measurement of anthocyanin content may have affected the anthocyanin stability of the genotypes under study. Numerous studies have shown that harvesting and storage time, temperature, pH, light and oxygen, degradation reactions during storage and processing influence anthocyanin stability [3, 72–75].

The total phenol concentration measured for the genotypes ranged from 269 to 7332 mg L$^{-1}$. The highest value was found in the genotype 67 (7332 mg L$^{-1}$) from Kerman province

followed by 60 and 128 originating Isfahan and Fars, respectively, which is higher than those reported in other studies. Collections of promising genotypes with high phenolic compositions can be destined for the production of juices [76]. Various studies on the chemical compounds of pomegranate genotypes/cultivars cultivated in different countries have also reported different amounts of total phenol in aril juice of pomegranates [77–80]. Variability in total phenol content of pomegranate in different studies can be related to genotype, environmental conditions, extraction method, and maturity [67, 80–82]. Since colorless polyphenols act as major compounds in the biological activities of pomegranate [2], information on the amount of phenol in pomegranate juice of different genotypes can help us in choosing genotypes for the juice and pharmaceutical industry.

Results showed a wide range of antioxidant activity (AA) in the Iranian genotypes studied, with values ranging between 5.23 and 97.82%. Genotypes that had more than 75% antioxidant activity belonged to Fars and Isfahan provinces. Diversity in the antioxidant activity of pomegranate juice can be attributed to fruit maturation, agricultural factors, and especially, genetic differences [10, 37, 54, 76].

Total soluble solids of the genotypes ranged from 8.8 to 19.27%. Since the number of genotypes examined in this study was high, TSS had a wider range than the values mentioned for this parameter in other articles. In only one genotype belonging to Bushehr province, higher TSS than 18% was obtained, whereas in the rest genotypes lower TSS content was found. However, the values of most genotypes were similar to the range observed in pomegranate genotypes grown in Greece [83], Turkey [84], and California [85]. The amount of these compounds is influenced by genetic diversity as well as climatic conditions, for example, higher temperatures lead to higher.

Acidity content (expressed as citric acid content) varied from 0.46 to 2.2 g L$^{-1}$. The content of citric acid is the main composition of acidic taste in pomegranate fruits [69]. Studies of Tehranifar et al. [35] and Fadavi et al. [86] on some Iranian pomegranate genotypes as well as the study of Caliskan and Bayazit [64] on 76 pomegranate accessions from Turkey also reported similar results to our research.

Pomegranate with an acidity content of less than 1.8% and a maturity index (MI) between 7 and 12 is suitable for fresh eating, and those with an MI between 11 and 16 are very tasty [87]. Accordingly, all but 3 of the studied genotypes had an acidity below 1.8% and 51 genotypes are suitable for the fresh market, 74.5% of which are quite delicious. Also, According to Martinez et al. [20] classification, most of the studied genotypes have a sour-sweet taste.

The pH values varied from 2.14 to 4.02, which agrees with the results reported by Hmid et al. [67]. The pH values reported by Tehranifar et al. [35] for 20 pomegranate cultivars in Iran ranged 3.16–4.09. Legua et al. [88] reported that the pH of six pomegranate cultivars was variable between 3.97 and 4.7.

## Correlation between targeted metabolites and morphological traits

The analysis of Spearman's coefficient of correlation between morphological and metabolite traits was showed that antioxidant activity correlated positively with total phenol and acidity, but correlated negatively with TSS (Fig 4). The genotypes whose fruit juices contain more phenolic compounds, lower total soluble solids, and more acidic pH have higher antioxidant activity. A positive correlation between antioxidant activity and phenolic compounds content was reported in previous studies [54, 89, 90].

It was found that the JA had a positive relationship with FFS. The values of pH correlated negatively with A and TSS (Fig 4). Khadivi-Khub et al. [29] also reported a negative correlation between titratable acidity and pH of 87 Iranian pomegranate accessions.

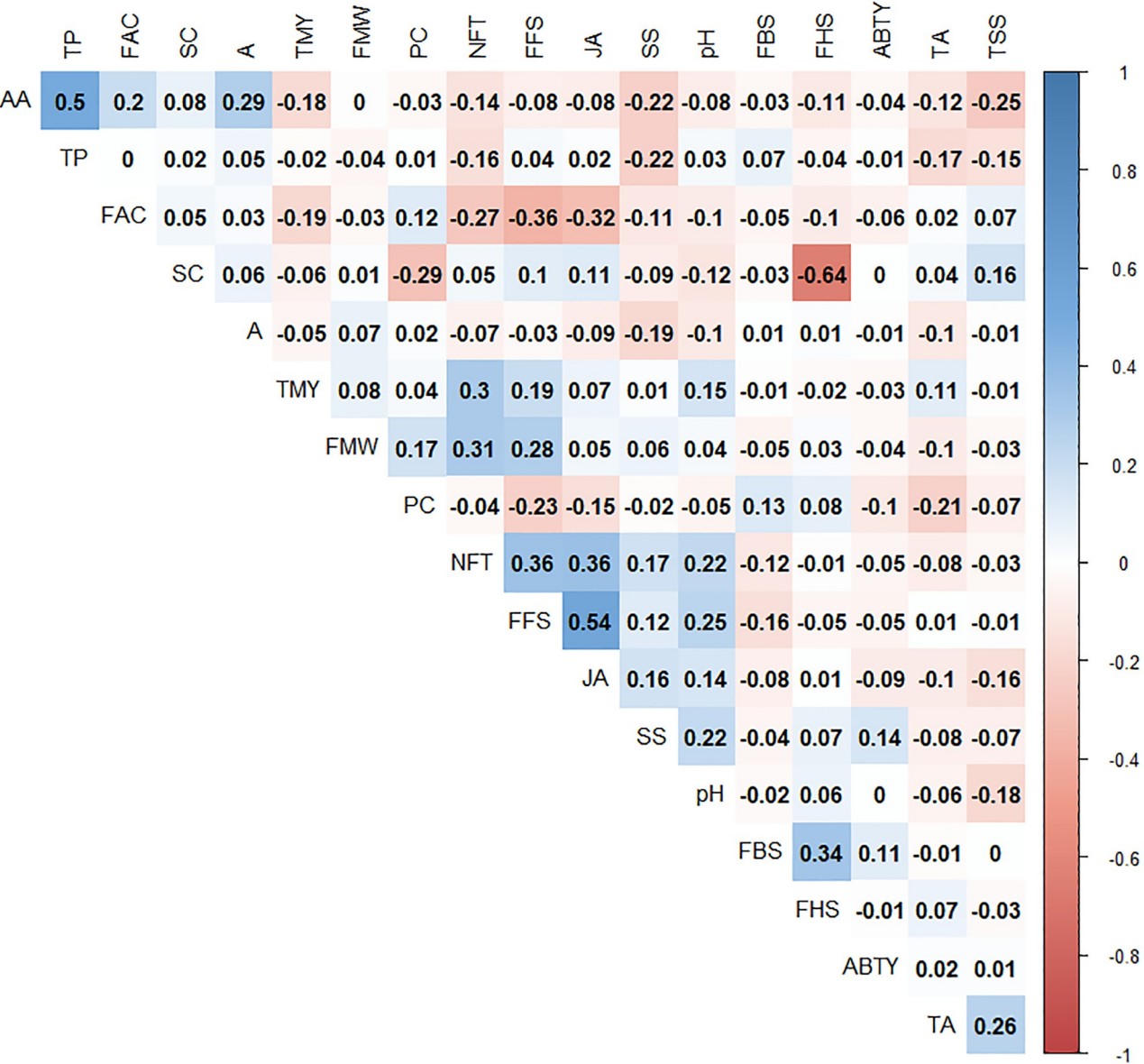

**Fig 4. Spearman correlation matrix of the traits.** Large circles display strong correlations and small circles represent weak correlations. Only significant correlations are shown (p value<0.05).

Anthocyanin content and the total phenol correlated negatively with each other (Fig 4). Reduction in the phenol content in some pomegranate genotypes might coincide with the increase in anthocyanin pigment content due to the contribution in the biosynthesis of the flavylium ring during the formation of anthocyanin [91, 92].

Fruitful flower size had a positive and significant correlation with the juiciness of aril, Average number of fruits per tree, Average fruit weight, and Average yield per tree (Fig 4). The larger the size of the fruitful flower, the higher the number of ovules, and consequently the higher percentage of fruit formation as well as better quality by the number of arils formed. Wetzstein et al. [93] reported that the percentage of fruit formation in large flowers was more than 95% compared with the smaller flowers which was less than 20%. The average fruit weight

increased significantly with increasing in flower size, which confirms the correlation of these two traits in this study. Therefore, the production of larger fruits requires the fertilization of thousands of ovules. In general, flower quality plays an important role in fruit production and the size of pomegranate fruit [93, 94].

Average yield per tree had a higher positive correlation with the number of fruits per tree than the weight of each fruit. It can be concluded that the tree yield depends more on the fruit number than the fruit weight. Wani et al. [22] showed there is a high positive correlation between the fruits number per tree and the yield of each tree within 33 pomegranate cultivars.

The 18 traits investigated in this study were classified into four distinct groups. Attributes AA, TP, A, FAC, and SC were included in the first group. Antioxidant activity had a positive correlation with other traits in this group. The third group consisted of five traits NFT, FFS, JA, SS, and pH that were positively correlated with each other. The three traits of TMY, FMW, and PC were also in another group, the fourth group included FB|S, FHS, ABTY, TA, and TSS.

## Principal component analysis and population structure of pomegranate germplasm

Principal component analysis (PCA) was carried out for 18 morphological and metabolic traits in all genotypes. The PCA recognized a total of eight components with Eigenvalues of more than one representing a cumulative 57.3% variability.

Principal component one (PC1) explained maximum variability of 14.9% of the total variation for traits and three morphological traits of FFS, NFT, and JA were the key characters for variability. The second component contributing 11.1% of the total variance has been influenced by characteristics of SC, FHS, and AA (Fig 5A). The third component explained 10.3% of the total variation. In general, the first three components accounted for 36.3% of the total variation.

Zarei [95] in evaluating different biochemical and pomological traits of 50 pomegranate accessions from Iran found that sees hardness, aril size, color-related properties, TA, antioxidant capacity, total phenol, and vitamin C were the most important for identification. Khadivi-khub et al. [29] also used PCA to evaluate 87 Iranian pomegranate local accessions based on morphological and chemical characters and reported the first three components had contributed 41.98% of the total variation. The results of PCA of 100 pomegranate genotypes from Iran using 40 morphological descriptors indicated that the first three components explained 22.24% of the variance [31] which was less than our study. Principal factor analysis of 25 morphological traits on 221 Iranian pomegranate genotypes classified traits in seven main groups which the first three principal components explained about 48.58% of the cumulative variance [33]. Among the traits used in that study, fruit skin color, fruit shape, flower position, and fruitful flower percentage appeared as the best traits to differentiate the pomegranate genotypes. Caliskan and Bayazit [64] evaluated the genetic diversity of 76 pomegranate accessions from Turkey based on morpho-pomological and biochemical characteristics and found the first three components had contributed 50% of the total variation. The PCA result of Dandachi et al. [96] on 78 pomegranate Lebanese accessions using 38 morphological and chemical descriptors showed that PC1, PC2, and PC3 accounted for 19.62%, 12.66%, and 9.1% of the variance, respectively (in total, 41.49% of the variance). They stated that sugar/acid ratio, fruit weight, and size are the most important traits in discriminating the studied accessions.

The genotype-by-trait biplot on the first two PCA axes was performed on all genotypes using morphological characteristics and metabolite content. It showed the relationships between the traits among the 152 genotypes. The cosine of the angles between the vectors shows the degree of correlation between the traits so that the acute and obtuse angles display

(A)

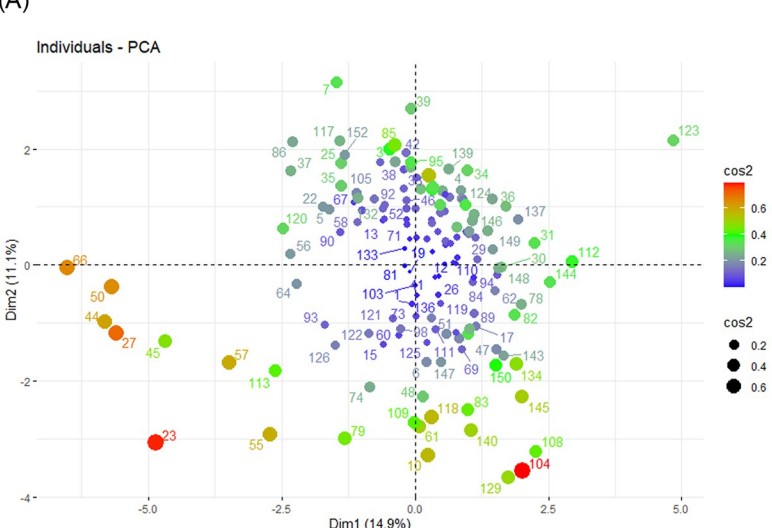

(B)

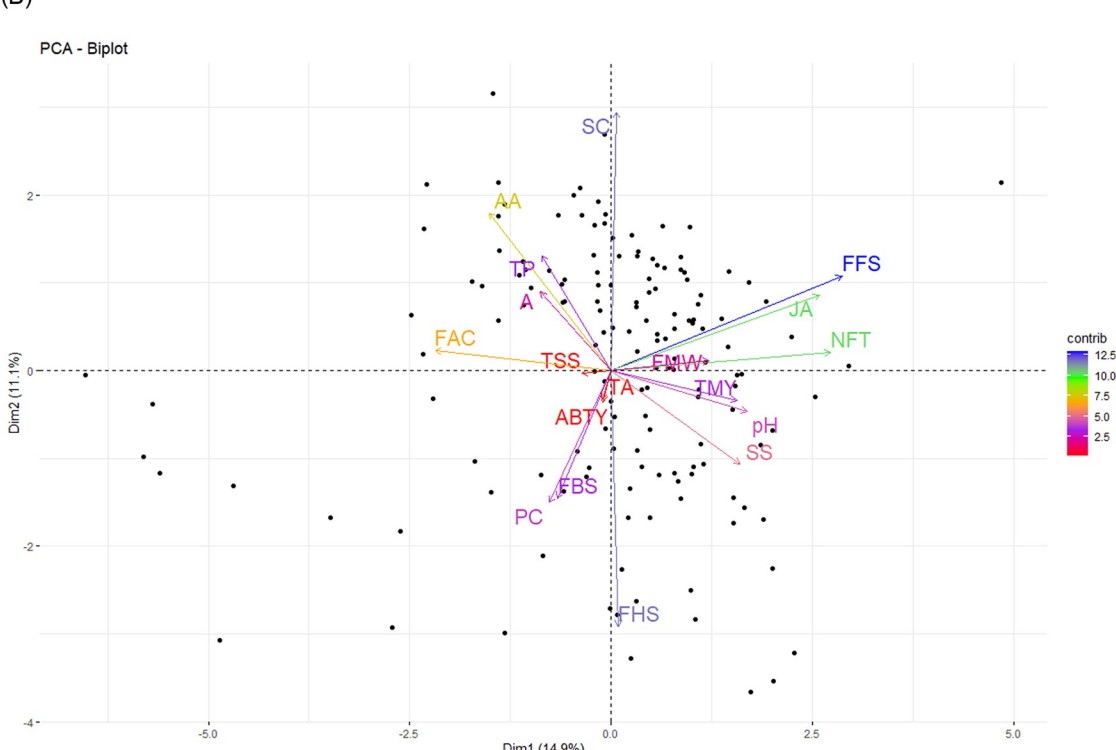

**Fig 5. Biplot of the first two principal components (PCs) for the studied pomegranate genotypes (A) and targeted metabolites and morphological traits (B).**

the positive and negative correlations, respectively. Accordingly, the variable FFS had a positive correlation with the JA, FMW and NFT, and a negative correlation with FAC. The values of AA correlated positively with TP, and A. These results corresponded to the results of Spearman correlation coefficient matrix presented in Fig 5B.

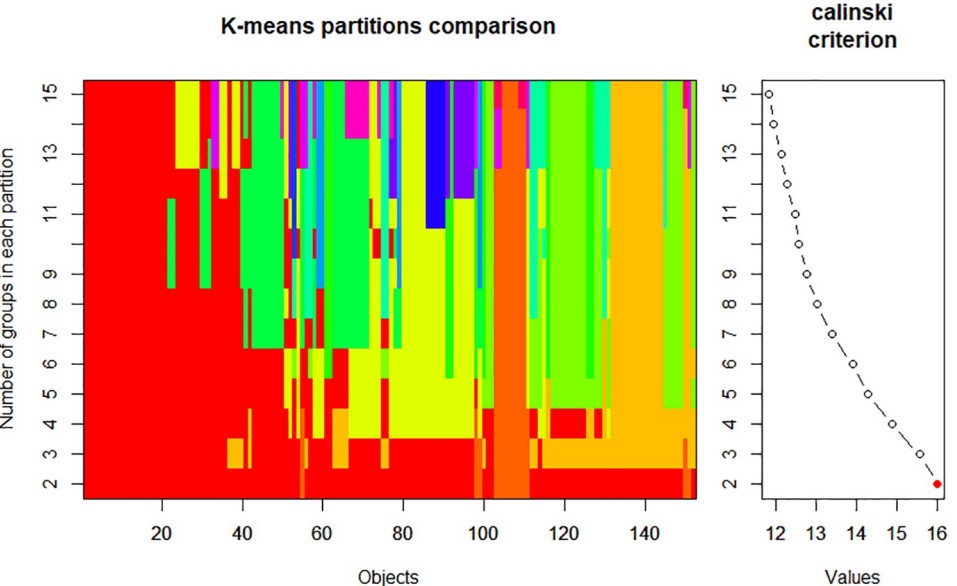

**Fig 6. Population structure of 152 pomegranate genotypes based on 18 morphological and biochemical characteristics (A), and determination of the optimal number of groups (k) using Calinski-Harabasz criterion.**

The biplot of pomegranate genotypes also display that the similar genotypes are grouped together on the plot. Due to the high diversity found among the genotypes, their distribution on the biplot of the first two principal component axes did not have an obvious grouping based on the measured traits. However, the genotypes with the smallest fruitful flower and arils with the least juicy were located at the lower-left part of the biplot. Based on the squared Cosinus values, genotypes 23, 108, 27, and 50 are good representatives of the important traits contributing to the first two principal components (Fig 5A).

One hundred fifty-two pomegranate genotypes were assigned to two subgroups according to the Calinski-Harabasz criterion (CHC) values. As shown in Fig 6, the highest CHC value was observed in K = 2.

The first cluster comprised 138 genotypes, and group 2 contained 14 genotypes. The genotypes with the smallest fruitful flowers and the least aril juicy were distributed in group 2 and originated from seven different provinces of Iran (Fig 7). Therefore, the grouping had no relationship with the origin of genotypes.

Cluster analysis classified the 152 genotypes into four main clusters (Fig 8). The first cluster contains 35 genotypes with the least NFT. The second cluster was mainly composed of 32 genotypes with white to pinkish-white seed colors. Genotypes with the highest levels of TA were placed in the fourth group. The third cluster enclosed the remaining 45 genotypes.

Population structure analysis of 738 Iranian pomegranate accessions with data of SSR markers was revealed eight groups that did not correspond to the geographical origin of the accessions [26], which is in agreement with the present results. Razi et al. [33] evaluated the morphological characteristics of 221 Iranian pomegranate genotypes, and the genotypes were divided into three subpopulations according to the K-means partitioning and cluster analysis.

## Core collection construction

The 152 genotypes selected came from 13 different provinces of Iran (two genotypes with an unknown origin), the largest number of which originated from Fars, followed by

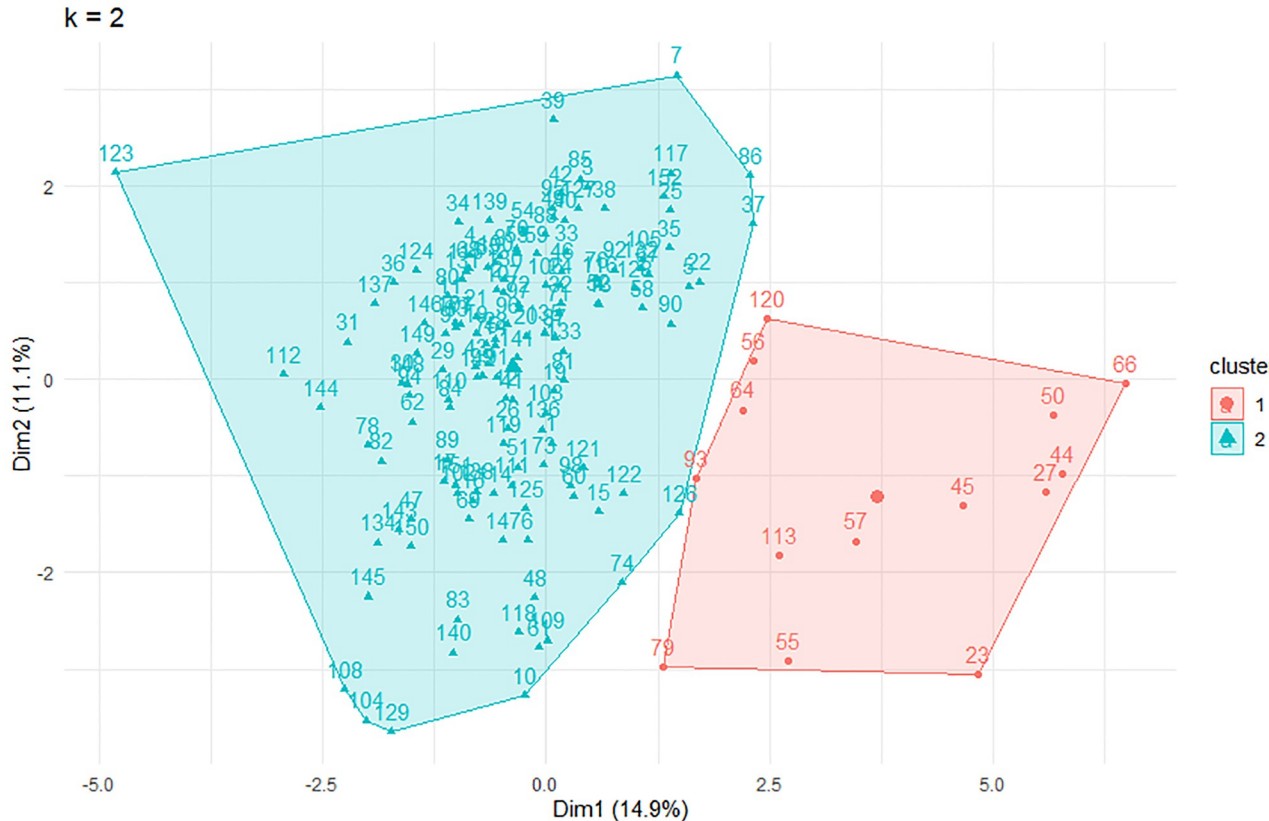

**Fig 7. Biplot representation of K-means clustering using the first two principle components for 152 pomegranate genotype (numbers) according morphological and chemical traits.**

Sistan and Baluchestan, Isfahan and, Yazd accounted for approximately 60% of the total collection.

From the total collection, 20 genotypes (13.16%), representing eight provinces were selected for a core set using maximization strategy through a modified heuristic algorithm (Fig 9).

According to Brown [97] and Diwan et al. [98] that the size of the core collection should be about 10% of the entire collection, while maintaining at least 70% of the genetic diversity of the initial set, the core collection in the present study is of appropriate size. However, Li et al. [99] stated that the size of the core subset should change depending on the size of the original collection. Previously developed pomegranate core collection size by Razi et al. [33] with 25 morphological traits was 5.6% of the entire collection (225 genotypes) which is lower in size than the core collection obtained in this study. Kazemi et al. [26] also developed a core collection of pomegranate based on the data of 12 SSR markers on 738 accessions, which comprised 34 accessions (about 4.61% of the total collection).

In order to evaluate the diversity in the core collection formed based on the morphological and biochemical traits compared to the original collection, four parameters were calculated. The mean difference (MD) between the core collection and the initial collection was 7.98%. The value of the coincidence rate of range (CR = 100%) showed a homogeneous distribution of traits in the core collection. Core collections with a CR greater than 80% and MD less than 20% are recognized as a suitable set for breeding purposes (Hu et al. 2000). Accordingly, the core collection constructed from Iranian pomegranate germplasm could be considered to

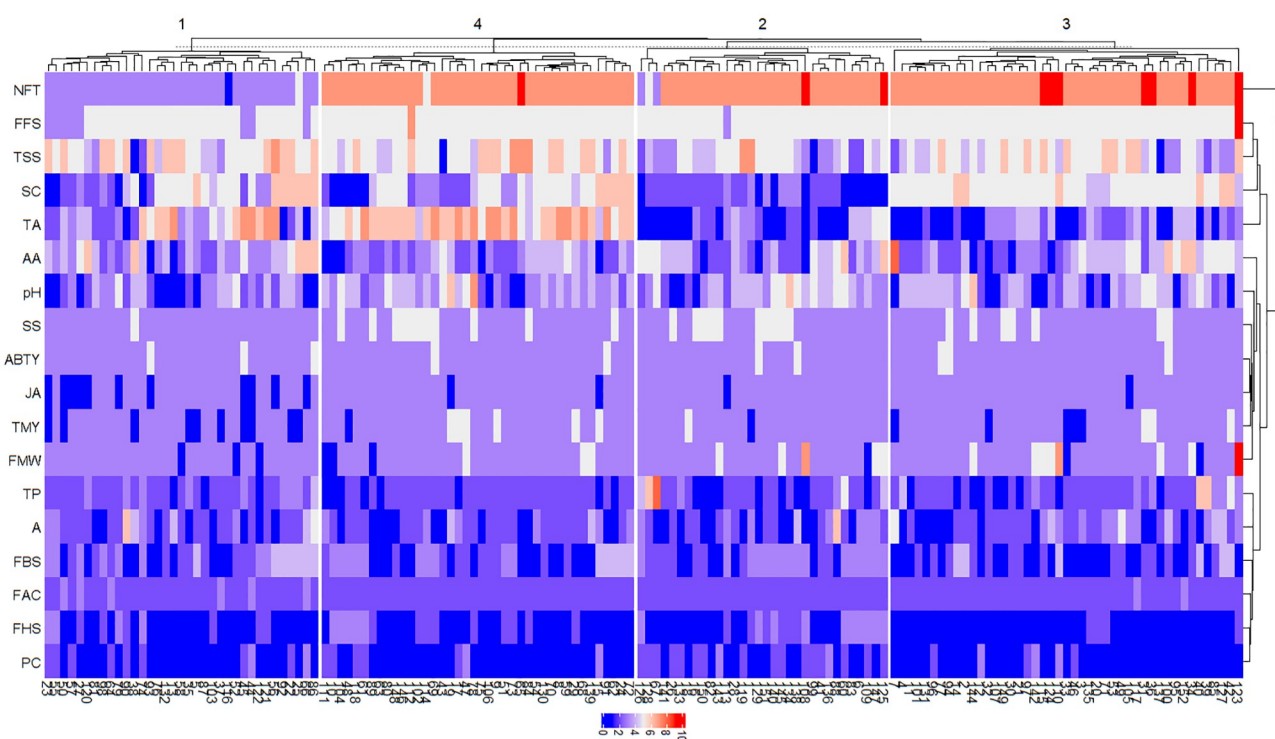

**Fig 8. Metabolic and morphological traits heatmap of the 152 pomegranate genotypes based on Euclidean distances.**

represent the genetic diversity in the initial collection. In addition, the variance difference (VD) and variance variable rate (VR) coefficient between the core and primary collection were calculated 33.9 and 122.96%, respectively. The higher values in VR and VD also indicate a more effective core collection [43]. The value VD in the present study indicated that there is a good variation between genotypes within the core collection. In conclusion, the core collection maintained the genetic diversity of the original population.

In the present study, the Shannon-Weaver diversity index values for the most morphological and metabolic traits were approximately similar in the primary set and core collection (Table 3), indicating that the variability of the total collection was represented in the core sub-set. Nevertheless, the Shannon-weaver index mean of the pomegranate core collection was greater than that of the entire collection. This may be due to the elimination of genetic redundancy in the core subset compared to the initial set. This reveals that the diversity in the core collection has increased. In the core collection established from the germplasm of the *Perilla frutescens* L. was also observed that the averages of Shannon and Nei diversity indices in the Core collection were higher than those in the total collection [100]. In can be concluded that the constructed core collection facilitates experiments to evaluate germplasm under different environmental conditions and the breeding programs in the future.

## Modeling the geographical distribution of pomegranate

It is necessary to determine which areas in the country are suitable for the establishment of pomegranate orchards as well as the construction of sites for the maintenance of the core collection for research purposes. The data in this study and bioclimatic variables were used to create a climate suitability map for pomegranate cultivation in Iran using the MaxEnt model.

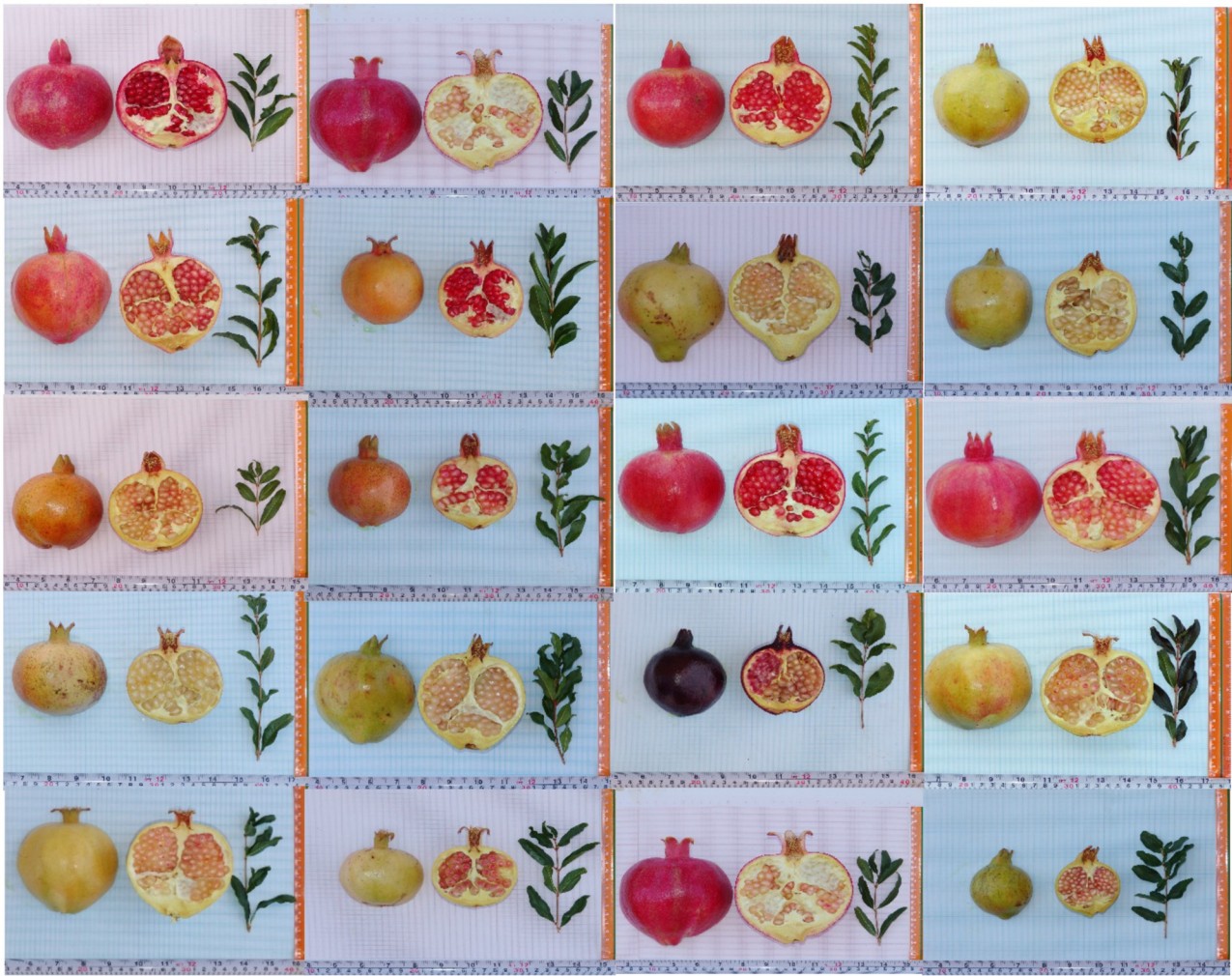

**Fig 9. Fruits of the selected pomegranates genotypes for a core collection.**

**Model evaluations and important environmental variables.** The ROC curve method was used to assess the performance of the MaxEnt model. The average test AUC (area under ROC curve) value was 0.938 which indicated the high accuracy of the model for predicting the potential geographical distribution areas of pomegranate during the current period (Fig 10). Values of AUC were used to verify the accuracy of MaxEnt models by the researchers, who stated that the values above 0.9 indicate the model's excellent performance in predicting species distribution [63, 101].

Relative contributions of the environmental variables were estimated using the MaxEnt model. The factors of altitude (alt), mean temperature of the coldest quarter (Bio 11), isothermality (Bio 3), mean temperature of the warmest quarter (Bio 10), mean temperature of the wettest quarter (Bio 8), and mean temperature of the driest quarter (Bio 9) with contribution rates of 18.9%, 12%, 11.9%, 8.9%, 7.8%, and 6.3%, respectively, were major factors affecting the habitat distribution of pomegranate. The cumulative contributions of these variables reached 65.8%. The permutation of the environmental variables in the model indicated that alt (29.3%), precipitation of coldest quarter (Bio 19; 17.7%), precipitation of driest month (Bio 14;

**Table 3. Shannon-Weaver diversity index for morphological and biochemical traits in the core and entire collection of pomegranate.**

| Trait | Core collection | Entire collection |
|---|---|---|
| Number of fruit per tree | 1.26 | 0.89 |
| Mean fruit weight | 0.89 | 0.62 |
| Mean tree yield | 0.69 | 0.58 |
| Anthocyanin in branch of this year | 0.42 | 0.24 |
| Petiole color | 0.69 | 0.72 |
| Fruitful flower size | 0.71 | 0.30 |
| Fruit albedo color | 0.20 | 0.21 |
| Fruit bottom shape | 1.29 | 1.31 |
| Fruit heel shape | 0.90 | 0.80 |
| Juice of aril | 0.42 | 0.32 |
| Seed size | 0.61 | 0.49 |
| Seed color | 1.71 | 1.60 |
| Total anthocyanin | 1.85 | 1.89 |
| Antioxidant activity | 1.85 | 1.70 |
| Total phenol | 1.51 | 1.10 |
| Acidity | 1.64 | 1.33 |
| Total soluble solids | 1.78 | 1.55 |
| pH | 1.75 | 1.62 |
| Average | 1.12 | 0.93 |

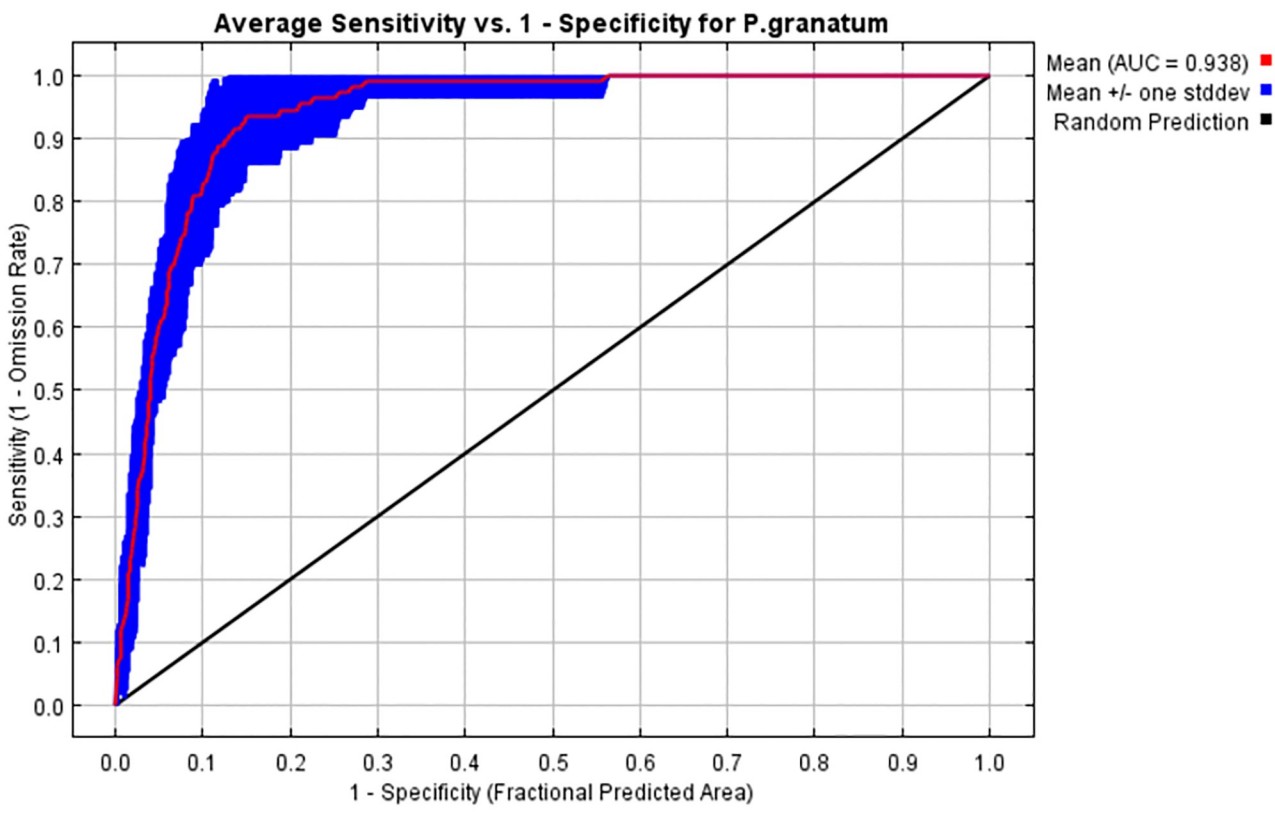

**Fig 10. ROC curve of test data to model habitat distribution of pomegranate under the current period.**

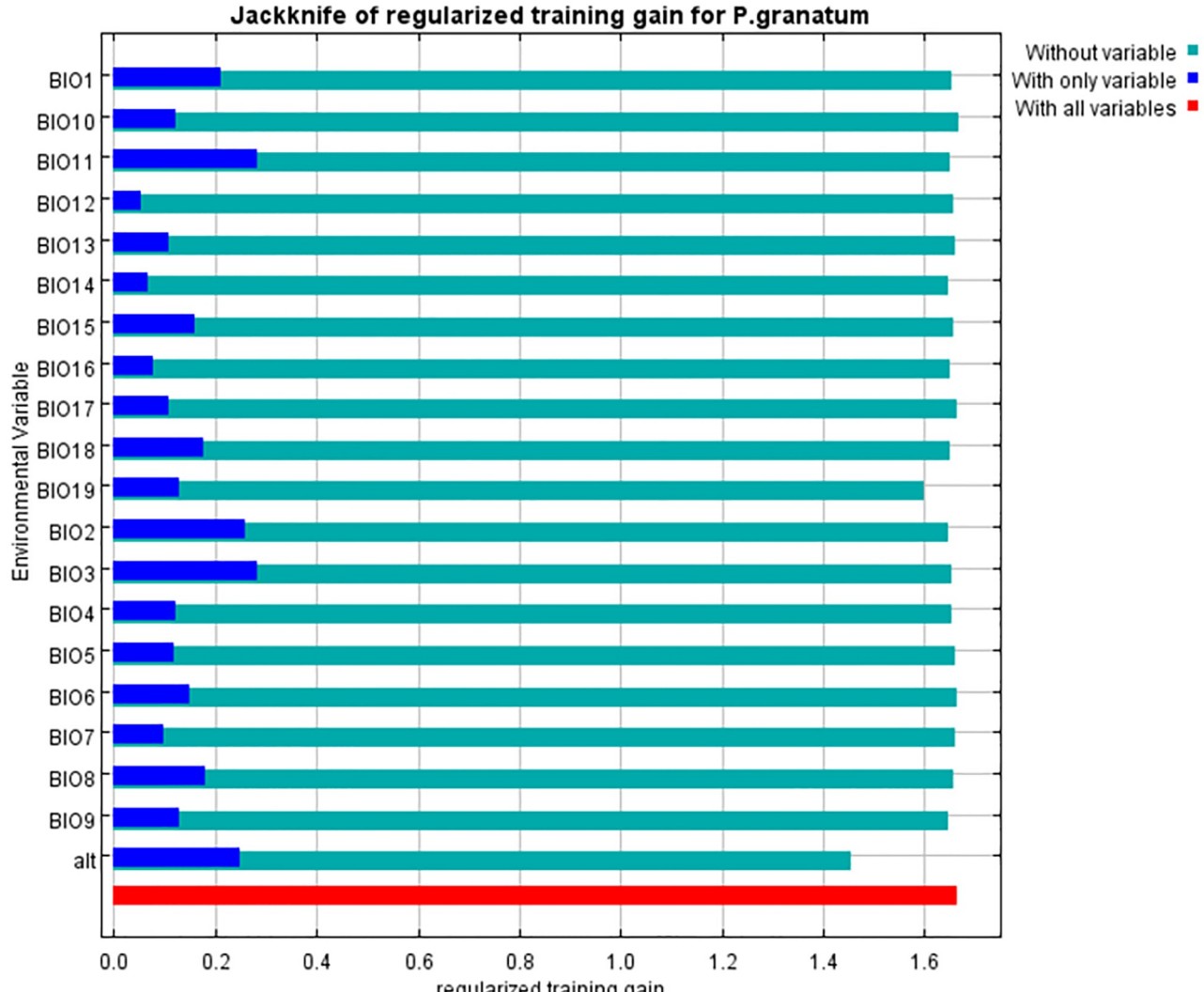

**Fig 11. Jackknife test for evaluating the relative importance of climate variables on the distribution of pomegranate using training gain.** Values are average on 10 replicate MaxEnt runs.

10.8%), and precipitation of wettest quarter (Bio 16; 10.8%) played main roles in predicting the potential distribution of pomegranate.

To get alternate estimates of variable importance, a jackknife test has been done (Fig 11). This test revealed that the variables Bio 3, Bio 11, Bio 2 (mean monthly temperature range), altitude, and Bio 1 (annual mean temperature) had a relatively higher contribution to the model. In contrast, annual precipitation (Bio 12) had the least impact on the distribution of pomegranate. Based on these results, temperature conditions play a more important role than rainfall in the distribution of pomegranate and it seems that pomegranate prefers warm and dry environments. However, omitting each variable (except for altitude) turning to the light blue bar did not decrease the training gain substantially. So, it appears that no variable contains a considerable amount of useful information that is not already contained in the other variables.

The thresholds (presence probability > 0.2) for the main variables were obtained using the response curves (Fig 12). The response curves indicate the quantitative relationship between

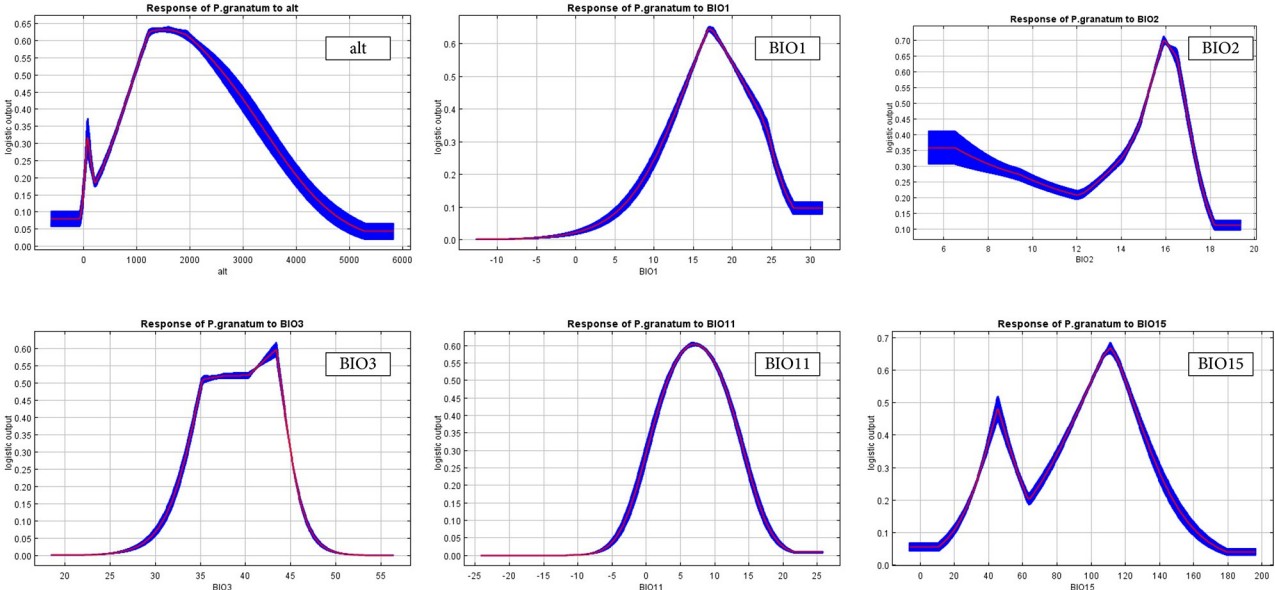

**Fig 12. Response curves for important environmental variables in the species distribution model for pomegranate.** Red curves shows the mean response of the 10 replicates, and blue shade is the mean +/- one standard deviation.

the environmental variables and the logistic probability of habitat suitability. According to these, isothermality (Bio3) ranged from 32.5 to 43.5%. The suitable habitat occurs when the mean temperature of the coldest quarter (Bio 11) range from -1.2 to15.8°C with a peak at 7°C. The mean monthly temperature (Bio2) range was 5–17.6°C, the annual mean temperature (Bio 1) ranged from 9 to 26°C, and the probability of pomegranate presence was decreased when the value of this variable was greater than 17°C. The altitude (alt) range varied from 250 to 4000 m with an optimal elevation at 1300–1800 m. The range of precipitation seasonality (Bio 15) was 29 to 148.5 mm. Areas with a temperature of coldest quarter greater than 19°C and lower than -5°C, and with an elevation higher than 4500 m were not suitable for pomegranate (Fig 12). It can be concluded that the most suitable zones for pomegranate growth are areas with an altitude of 1300–1800 m, the annual mean temperature of 17°C, and the mean temperature of the coldest quarter of 7°C

## Potential distribution of pomegranate under current climate

Current potential distribution areas of pomegranate occurred in most parts of Iran except highlands of northwestern of the country, Zagros Mountain Range, North Khorasan and the northern parts of Razavi Khorasan, the northern part of Tehran province, and the southern parts of provinces of Hormozgan and Sistan and Baluchestan.

The areas with the highest suitability were mainly located in the provinces of Yazd, South Khorasan, Semnan, Isfahan, south of Razavi Khorasan, Fars and Kerman, which were mostly located in central and eastern Iran (Fig 13) which are corresponding to the main areas of pomegranate cultivation.

The northwestern regions of the country (especially Ardabil), the northern parts of Razavi Khorasan, the northern part of Tehran province, and the north of Tehran province (highlands located in the Zagros and Alborz Mountain Range) have a cold mountain climate with cold winters and cool summers compared to other regions of Iran. The southern parts of Sistan and Baluchestan, and Hormozgan provinces along the Oman Sea and the Persian Gulf have a

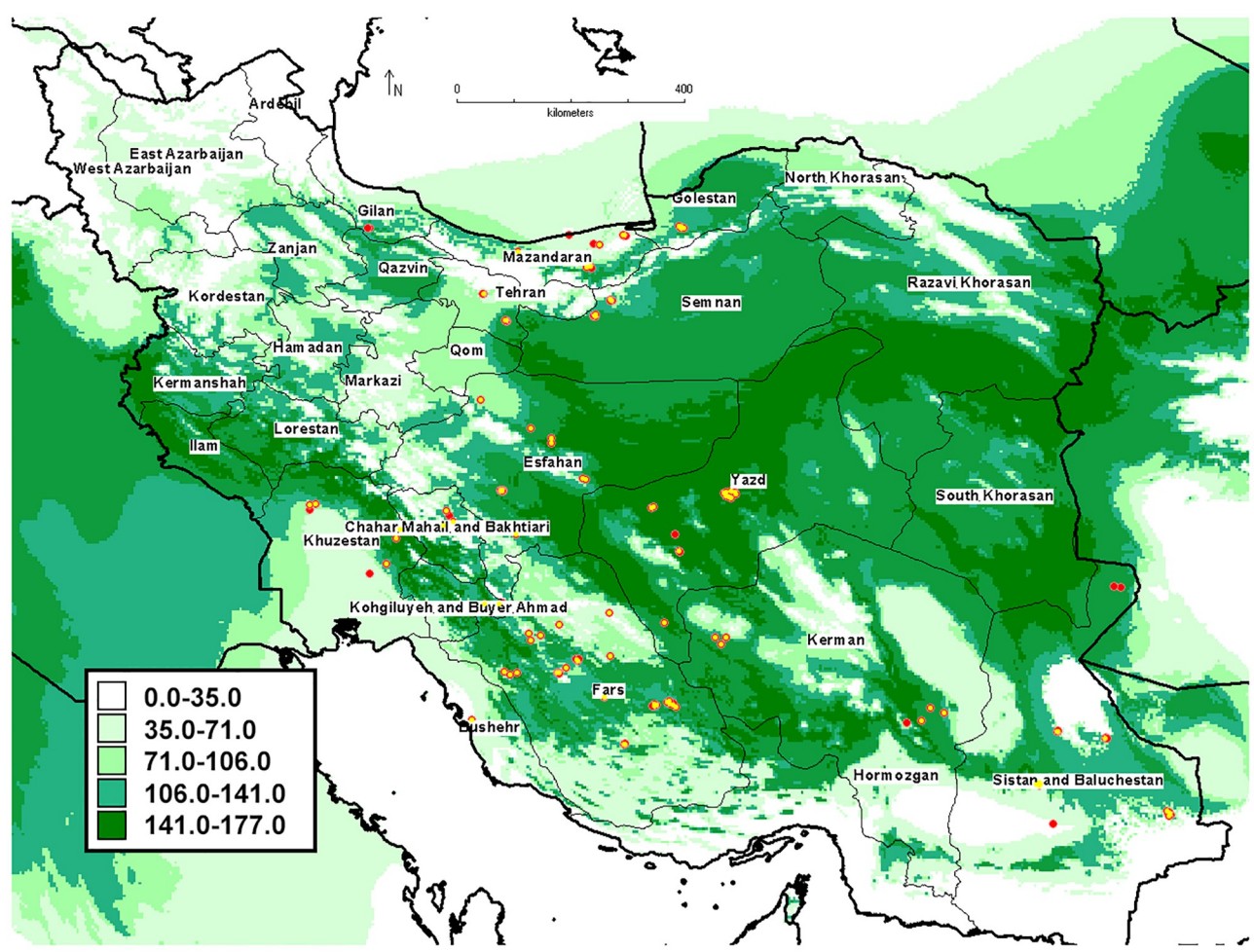

**Fig 13. Current spatial distribution of pomegranate in Iran.**

coastal dry climate with hot and humid summers [102]. The climatic conditions of these areas are not favorable for the growth of pomegranate fruit.

Although pomegranate has adapted to the climatic conditions of different regions of Iran, favorable climatic conditions for the growth of pomegranate fruit are offered to be Mediterranean-like climates, which include hot dry summers without precipitation during the last stages of the fruit's development, heavy sunlight, and cool winter. The tree is damaged at temperatures below -18˚C, but it can resist temperatures up to 45–48 ∘C [2]. Pomegranate also tolerates drought [103]. The highly suitable areas were mainly concentrated in the central and eastern regions of Iran have an arid and semi-arid climate to which pomegranate is adapted, and the distribution of pomegranate in these parts is reasonable.

## Potential distribution of pomegranate over the future period

Numerous studies have shown that the temporal and spatial pattern of rainfall and temperature in Iran will change in the coming decades. Mean temperatures in Iran will be increased by 2.6˚C in the next decades [104]. Ashraf Vaghefi et al. [102] predicted that the maximum temperature would rise by 1.1 to 2.5˚C throughout Iran. The country will experience a 30–

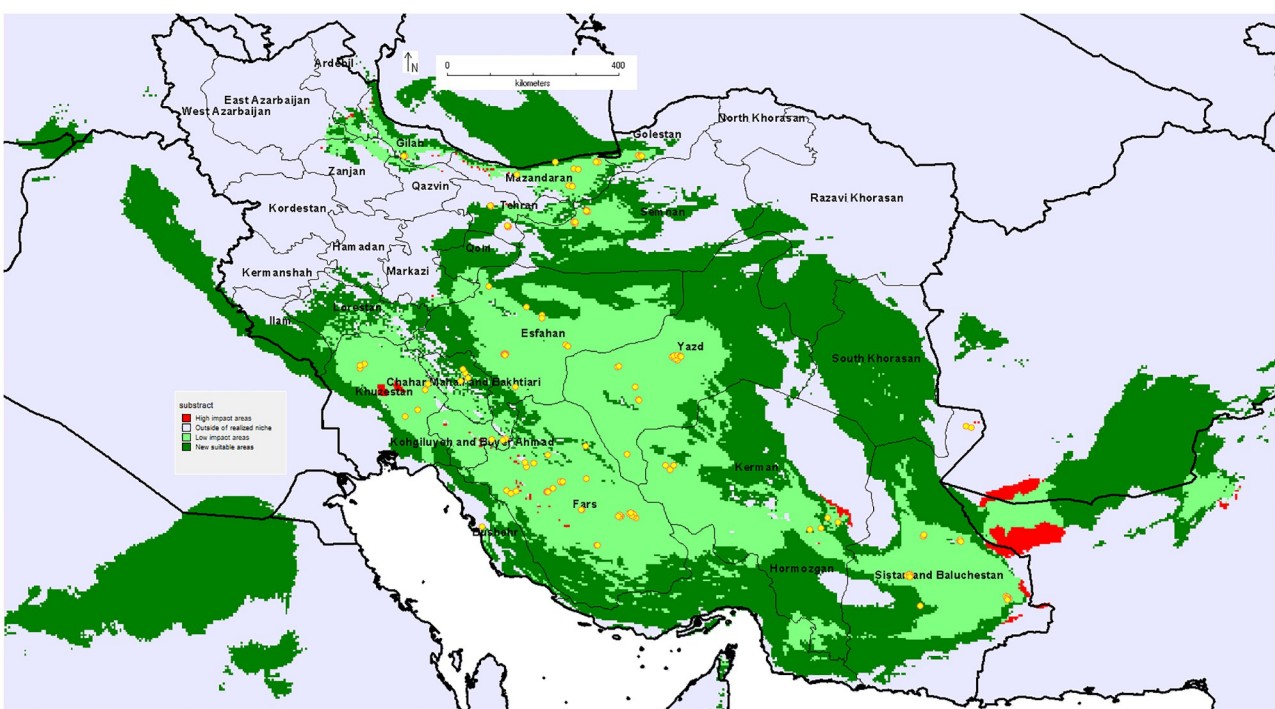

**Fig 14. Future potential spatial distribution of pomegranate in Iran under the CCM3 climate model.**

35% decline in annual rainfall [104, 105]. Hence, the potential future distribution of pomegranate in Iran was predicted using the MaxEnt model. Some researchers have used this model to predict the spatial distribution of tree species under future climatic conditions [49, 106, 107].

The predicted distribution map of pomegranate in the future showed that the main provinces of pomegranate cultivation would be less affected by future climatic conditions. Only part of Khuzestan province will be lost for pomegranate cultivation in the future.

According to the model, the northwestern and northeastern parts of the country will not be suitable for pomegranate cultivation in the future (Fig 14). considering that pomegranate is an endemic species of Iran and it is tolerant of heat and drought [103], it seems that climate change will not have much impact on its geographical distribution in the near future. In addition, the main pomegranate growing areas for commercial production are concentrated in the arid and semi-arid climates of Iran. As a result, pomegranate production is not expected to be affected by projected climate change until the middle of the century. The result of Hong-Qun et al. [108] also demonstrated that the potential geographical distribution of the *Taxus wallichiana* var. *mairei* was nearly similar in current and future climatic conditions.

New suitable climatic conditions have been predicted in Hormozgan province (Fig 14). The frequency of dry periods (for ≥120 consecutive days, rainfall <2 mm day−1 and Tmax ≥30˚C) according to the prediction of Ashraf Vaghefi et al. [102] will decrease for the middle of the century in the northwestern regions and parts of the northeast of the country, while increasing such periods are predicted for the rest of areas. Probably the decrease in humidity and the creation of semi-arid climatic conditions in Hormozgan province will provide the conditions for pomegranate growth in the near future.

## Conclusion

In the present study, a core collection was constructed for the first time on 152 Iranian pomegranate genotypes using a combination of morphological traits and targeted metabolites, which maintained the genetic diversity of the initial collection. Also, modeling the geographical distribution of pomegranate under current and future climatic conditions in Iran showed the suitable habitat areas for pomegranate were located in central parts of Iran with an arid and semi-arid climate conditions which had little effect on the distribution of this species until the middle of the century. According to the MaxEnt model, the three main variables affecting the habitat distribution of pomegranate included altitude, mean temperature of coldest quarter, and Isothermality. The present results indicated that the highest metabolite content was observed in some genotypes of Kerman and Fars, while the highest yield related to those of Sistan and Baluchestan and Fars provinces that desired genotypes could be selected depending on the commercial or pharmaceutical production purposes. However, core collection genotypes should be grown in different climatic conditions according to the effect of environmental variables on the yield and metabolic content of genotypes to finally select the superior genotypes for different climatic regions. Overall, the results of this study provide information for use in breeding programs for the production of new pomegranate cultivars with high nutritional quality as well as identification of high suitable agro-ecological zones for the establishment of orchards to improve yield and metabolic properties of pomegranate fruits.

## Supporting information

**S1 Table. Morphological traits and targeted metabolites of 152 pomegranate genotypes.** (DOCX)

## Author Contributions

**Conceptualization:** Mehrshad Zeinalabedini.

**Data curation:** Mehrshad Zeinalabedini, Mohammad Reza Vazifeshenas.

**Formal analysis:** Mehrshad Zeinalabedini.

**Funding acquisition:** Mehrshad Zeinalabedini.

**Investigation:** Maryam Farsi, Mohammad Reza Vazifeshenas.

**Methodology:** Mehrshad Zeinalabedini, Mohammad Reza Vazifeshenas.

**Project administration:** Mehrshad Zeinalabedini.

**Resources:** Maryam Farsi.

**Software:** Mehrshad Zeinalabedini.

**Supervision:** Mansoor Kalantar, Mehrshad Zeinalabedini, Mohammad Reza Vazifeshenas.

**Validation:** Maryam Farsi, Mehrshad Zeinalabedini.

**Visualization:** Mehrshad Zeinalabedini.

**Writing – original draft:** Maryam Farsi.

**Writing – review & editing:** Mansoor Kalantar, Mehrshad Zeinalabedini, Mohammad Reza Vazifeshenas.

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
