## [Decision Letter · Decision Letter 0]

6 Jun 2022

PONE-D-22-06916First assessment of Iranian pomegranate germplasm using targeted metabolites and morphological traits to develop the core collection and modeling of the current and future spatial distribution under climate change conditionsPLOS ONE

Dear Dr. Zeinalabedini,

Thank you for submitting your manuscript to PLOS ONE. After careful consideration, we feel that it has merit but does not fully meet PLOS ONE’s publication criteria as it currently stands. Therefore, we invite you to submit a revised version of the manuscript that addresses the points raised during the review process.

We look forward to receiving your revised manuscript.

Kind regards,

Mehdi Rahimi, Ph.D.

Academic Editor

PLOS ONE

Journal Requirements:

I have read the journal's policy and the authors of this manuscript have the following competing interests

7. We note that Figures 1-3 and 13-14 in your submission contain [map/satellite] images which may be copyrighted. All PLOS content is published under the Creative Commons Attribution License (CC BY 4.0), which means that the manuscript, images, and Supporting Information files will be freely available online, and any third party is permitted to access, download, copy, distribute, and use these materials in any way, even commercially, with proper attribution. For these reasons, we cannot publish previously copyrighted maps or satellite images created using proprietary data, such as Google software (Google Maps, Street View, and Earth). For more information, see our copyright guidelines: http://journals.plos.org/plosone/s/licenses-and-copyright.

a. You may seek permission from the original copyright holder of Figures 1-3 and 13-14 to publish the content specifically under the CC BY 4.0 license.  

Additional Editor Comments:

Dear Authors

Below, please find the reviewers' comments for your paper. You have carried out the corrections as suggested by the reviewer(s).

with Thanks

Reviewers' comments:

Reviewer's Responses to Questions

**Comments to the Author**

1. Is the manuscript technically sound, and do the data support the conclusions?

Reviewer #1: Yes

Reviewer #2: Yes

2. Has the statistical analysis been performed appropriately and rigorously? 

Reviewer #1: Yes

Reviewer #2: No

3. Have the authors made all data underlying the findings in their manuscript fully available?

Reviewer #1: Yes

Reviewer #2: Yes

4. Is the manuscript presented in an intelligible fashion and written in standard English?

Reviewer #1: Yes

Reviewer #2: Yes

5. Review Comments to the Author

Reviewer #1: Manuscript titled:

First assessment of Iranian pomegranate germplasm using targeted metabolites and morphological traits to develop the core collection and modeling of the current and future spatial distribution under climate change conditions

There are several linguistic and typographical mistakes in this paper that must be corrected.

1. What is novelty in the abstract? It needs improvement

2. Reference should be according to journal format

3. The manuscript must revise with native English speakers because the manuscript contains a lot of mistakes in grammar and structure, for example, Line 302

“The analysis of Spearman coeﬃcient of correlation between morphological and metabolite traits was showed antioxidant activity correlated positively with total phenol”. Rephrase sentence.

4. Line 312. “also reported there was a negative correlation”. Rephrase the sentence.

5. The discussion section is just descriptive of the results, there are no logical reasoning. Give some recent and comprehensive literature in discussion and support the obtained results. Do not give simple biological inference.

6. Do add the following latest references in discussion part for justification of your results, please.

Zarei A (2017). Biochemical and pomological characterization of pomegranate accessions in fars province of Iran. SABRAO J. Breed. Genet. 49(2): 155–167.

Tarinta T, Chanthai S, Lertrat K, Nawata E, Techawongstien S (2020). Identification of the secondary metabolite capsiate in Capsicum germplasm accessions. SABRAO J. Breed. Genet. 52(2):144–157.

Upadyshev MT (2022). Apple cultivars and rootstocks assay for the identification of diverse viruses and healthy genotypes for breeding. SABRAO J. Breed. Genet. 54(1): 79-87. http://doi.org/10.54910/sabrao2022.54.1.8.

Motyleva SM, Medvedev SM, Morozova NG, Kulikov IM (2021). Leaf micromorphological and biochemical features of scab disease in immune and moderately resistant columnar apple (Malus domestica) cultivars. SABRAO J. Breed. Genet. 53(3): 352-366.

Hong-Qun, L. I., Li-Gang, X. I. N. G., & Xie-Ping, S. U. N. (2022). Predicting the potential distribution of Taxus wallichiana var. Mairei under climate change in China using maxent modeling. Pak. J. Bot. 54(4): 1305-1310.

Ahmad, N., Shakil, A., Shinwari, Z. K., Ahmad, I., & Wahab, A. (2022). Phytochemical study and antimicrobial activities of extracts and its derived fractions obtained from Berberis vulgaris L. and Stellaria media L. leaves. Pak. J. Bot. 54(4): 1517-1521.

Regards

Reviewer #2: - In Materials & Methods, authors explained that 152 genotypes selected from 760 genotypes based on the results of Kazemi alamuti et al. [26]. It is better to explain in brief how and why these genotypes selected and present the genotypes name, source/origin (if applicable) as a supplementary Table. I would appreciate to see official reference numbers for all accessions so that they can be order from reference gene banks and benefit for the entire pomegranate scientists community.

- In Materials & Methods, section “Investigation of morphological characteristics”: explain how many plants per genotypes evaluated? Some information about the field? Are there any statistical design? Or just sampled from a tree in a collection site? Please clarify it.

- Line 124, page 6: “The pH measurement was performed using a digital pH meter (Metrohm model) at 21 °C”, pH of what? Clearly explain.

- Fig 4. Spearman correlation matrix of the traits. This figure is difficult for reader to well understanding the relationship between traits. It is recommended to summarized in a Table.

- I strongly recommended to present a Table about some statistical parameters for all measured traits, like as mean, maximum, minimum, standard deviation and coefficient of variation (CV). This can be better explain the amount of variation in the studied germplasm.

- Are there any ANOVA analysis for traits?

6. PLOS authors have the option to publish the peer review history of their article (what does this mean?). If published, this will include your full peer review and any attached files.

Reviewer #1: **Yes: **PROF. DR. NAQIB ULLAH KHAN

Reviewer #2: **Yes: **Reza Talebi

---

## [Author Response · Author response to Decision Letter 0]

27 Nov 2022

Dear Roland Paile Bendaña

We would like to thank you for your email. I apologize for the delay in replying to your email. I could not send an email during this period due to the lack of internet access.

To resolve the issue mentioned in the email, all data are fully available without restriction, and we confirm that manuscript and Supporting Information files contain "minimal data set" to reach the conclusions drawn in the manuscript with related methods, and any additional data required to replicate the reported study findings in their entirety. 

Kind regards,

Mehrshad Zeinalabedini

Dear Dr. Mehdi Rahimi, 

We would like to thank you and the reviewers for constructive assessment of our manuscript entitled "First assessment of Iranian pomegranate germplasm using targeted metabolites and morphological traits to develop the core collection and modeling of the current and future spatial distribution under climate change conditions". We are pleased that all reviewers have found our article interesting. The authors greatly acknowledge the constructive comments made by the two reviewers. We have now revised the manuscript according to reviewers' comments, and a point-by-point response is submitted. Further, we confirmed that the funders had no role in study design, data collection and analysis, decision to publish, or preparation of the manuscript. Besides, the authors received no specific funding for this work.

We appreciate you for considering the revised version of the manuscript and look forward to receiving your decision.

Sincerely,

Mehrshad Zeinalabedini

Mehrshad Zeinalabedini, PhD

Agricultural Biotechnology Research Institute of Iran (www.abrii.ac.ir)

Mahdasht Road, Karaj, Iran. P.O. Box: 31535-1897 

Tel: +98 (261) 2703536 

Fax: +98 (261) 2704539 

Email: mzeinolabedini@abrii.ac.ir

Answers to editor comments

Ref: PONE-D-22-06916

Title: First assessment of Iranian pomegranate germplasm using targeted metabolites and morphological traits to develop the core collection and modeling of the current and future spatial distribution under climate change conditions

Journal: Plos One

Re: The format was checked and adjusted based on the editorial comments.

Re: This project was supported by the Agricultural Biotechnology Research Institute of Iran, ABRII07-05-05-92117. We revised this part based on the reviewer comment.

The funders had no role in study design, data collection and analysis, decision to publish, or preparation of the manuscript. At this time, please address the following queries:

Please clarify the sources of funding (financial or material support) for your study. List the grants or organizations that supported your study, including funding received from your institution. State what role the funders took in the study. If the funders had no role in your study, please state: "The funders had no role in study design, data collection and analysis, decision to publish, or preparation of the manuscript." If any authors received a salary from any of your funders, please state which authors and which funders. If you did not receive any funding for this study, please state: "The authors received no specific funding for this work." Please include your amended statements within your cover letter; we will change the online submission form on your behalf.

Re: we amended the requested statements in the cover letter based on the editorial comments.

I have read the journal's policy and the authors of this manuscript have the following competing interests

Re: we amended the requested statements based on the editorial comments.

Re: The data set used in this study was presented in Supplementary Table 1 (S1 Table).

Re: ORCID iD for Mehrshad Zeinalabedini as the corresponding author is 0000-0002-34364334. 

7. We note that Figures 1-3 and 13-14 in your submission contain [map/satellite] images which may be copyrighted. All PLOS content is published under the Creative Commons Attribution License (CC BY 4.0), which means that the manuscript, images, and Supporting Information files will be freely available online, and any third party is permitted to access, download, copy, distribute, and use these materials in any way, even commercially, with proper attribution. For these reasons, we cannot publish previously copyrighted maps or satellite images created using proprietary data, such as Google software (Google Maps, Street View, and Earth). For more information, see our copyright guidelines: http://journals.plos.org/plosone/s/licenses-and-copyright.

 a. You may seek permission from the original copyright holder of Figures 1-3 and 13-14 to publish the content specifically under the CC BY 4.0 license. 

Re: Figures 1-3 and 13-14 were designed by the free software, Diva-GIS (available at: http://www.diva-gis.org/climate; accessed on 4th October 2004) and MAXENT (https://biodiversityinformatics.amnh.org/open_source/maxent/) to construct the maps by the authors and had no copyright.

8. 

The authors have declared that no competing interests exist.

Answers to Reviewers comments

Reviewer #1

1. What is novelty in the abstract? It needs improvement. 

Re: Pomegranate core collection was established for the first time in Iran by combining morphological and metabolic data. Also, the impact of future climatic conditions (until the middle of the century) on the geographical distribution of pomegranate had not been investigated to date, which was predicted in this study. This information will help us identify the most suitable places for orchard construction in the future.

2. Reference should be according to journal format 

Re: References were formatted according to the NLM/ICMJE style (the journal format).

4. Line 302 “The analysis of Spearman coeﬃcient of correlation between morphological and metabolite traits was showed antioxidant activity correlated positively with total phenol”. Rephrase sentence.

Re: it was revised.

5. Line 312. “also reported there was a negative correlation”. Rephrase the sentence..

Re: The sentence was rephrased. 

 5. Do add the following latest references in discussion part for justification of your results, please.

Zarei A (2017). Biochemical and pomological characterization of pomegranate accessions in fars province of Iran. SABRAO J. Breed. Genet. 49(2): 155–167.

Tarinta T, Chanthai S, Lertrat K, Nawata E, Techawongstien S (2020). Identification of the secondary metabolite capsiate in Capsicum germplasm accessions. SABRAO J. Breed. Genet. 52(2):144–157.

Upadyshev MT (2022). Apple cultivars and rootstocks assay for the identification of diverse viruses and healthy genotypes for breeding. SABRAO J. Breed. Genet. 54(1): 79-87. http://doi.org/10.54910/sabrao2022.54.1.8.

Motyleva SM, Medvedev SM, Morozova NG, Kulikov IM (2021). Leaf micromorphological and biochemical features of scab disease in immune and moderately resistant columnar apple (Malus domestica) cultivars. SABRAO J. Breed. Genet. 53(3): 352-366.

Hong-Qun, L. I., Li-Gang, X. I. N. G., & Xie-Ping, S. U. N. (2022). Predicting the potential distribution of Taxus wallichiana var. Mairei under climate change in China using maxent modeling. Pak. J. Bot. 54(4): 1305-1310.

Ahmad, N., Shakil, A., Shinwari, Z. K., Ahmad, I., & Wahab, A. (2022). Phytochemical study and antimicrobial activities of extracts and its derived fractions obtained from Berberis vulgaris L. and Stellaria media L. leaves. Pak. J. Bot. 54(4): 1517-1521.

Re: Some of these articles were added in the results and discussion section, but we did not have access to the full article of some of them. As a result, we were unable to study them and add them to our article.

Reviewer #2: 

- In Materials & Methods, authors explained that 152 genotypes selected from 760 genotypes based on the results of Kazemi alamuti et al. [26]. It is better to explain in brief how and why these genotypes selected and present the genotypes name, source/origin (if applicable) as a supplementary Table. I would appreciate to see official reference numbers for all accessions so that they can be order from reference gene banks and benefit for the entire pomegranate scientists community.

Re: The genotypes in this study were selected based on the previous results of our different studies (Kazemi alamuti et al. 2012; Mousavi Derazmahalleh et al. 2013; Razi et al. 2021), which were performed on various objects e.g. genetic diversity, population genetic etc. of Yazd pomegranate national germplasm. The name and origin of genotypes also was presented in S1 Table based on the existing codes of Iranian pomegranate germplasm.

- In Materials & Methods, section “Investigation of morphological characteristics”: explain how many plants per genotypes evaluated? Some information about the field? Are there any statistical design? Or just sampled from a tree in a collection site? Please clarify it.

Re: The number of plants for evaluating each genotype was three individual from which eight samples were harvested.

In this collection, four trees per genotype have been cultivated in five blocks in an area of 19 hectares. Information on the geographical location and climatic conditions of the site of collection was mentioned in the article.

- Line 124, page 6: “The pH measurement was performed using a digital pH meter (Metrohm model) at 21 °C”, pH of what? Clearly explain.

Re: pH of pomegranate aril juice was measured. 

- Fig 4. Spearman correlation matrix of the traits. This figure is difficult for reader to well understanding the relationship between traits. It is recommended to summarized in a Table.

Re: Another figure, which includes the correlation coefficient values, replaced this figure. In the new figure, determining the correlation through both the values of the correlation coefficient and the color spectrum will make it easier for the reader to understand the correlation between traits.

- I strongly recommended to present a Table about some statistical parameters for all measured traits, like as mean, maximum, minimum, standard deviation and coefficient of variation (CV). This can be better explain the amount of variation in the studied germplasm.

Re: Descriptive statistics of biochemical traits (minimum, maximum, mean, and coefficient of variation) of 152 pomegranate genotypes were presented in Table 2.

The morphological features studied were qualitative traits that were scored and evaluated. 

- Are there any ANOVA analysis for traits?

Re: there was no ANOVA analysis for traits in this study.

---

## [Decision Letter · Decision Letter 1]

13 Dec 2022

PONE-D-22-06916R1First assessment of Iranian pomegranate germplasm using targeted metabolites and morphological traits to develop the core collection and modeling of the current and future spatial distribution under climate change conditionsPLOS ONE

Dear Dr. Zeinalabedini,

Thank you for submitting your manuscript to PLOS ONE. After careful consideration, we feel that it has merit but does not fully meet PLOS ONE’s publication criteria as it currently stands. Therefore, we invite you to submit a revised version of the manuscript that addresses the points raised during the review process.

We look forward to receiving your revised manuscript.

Kind regards,

Mehdi Rahimi, Ph.D.

Academic Editor

PLOS ONE

Journal Requirements:

Reviewers' comments:

Reviewer's Responses to Questions

**Comments to the Author**

1. If the authors have adequately addressed your comments raised in a previous round of review and you feel that this manuscript is now acceptable for publication, you may indicate that here to bypass the “Comments to the Author” section, enter your conflict of interest statement in the “Confidential to Editor” section, and submit your "Accept" recommendation.

Reviewer #1: (No Response)

Reviewer #2: All comments have been addressed

2. Is the manuscript technically sound, and do the data support the conclusions?

Reviewer #1: Yes

Reviewer #2: Yes

3. Has the statistical analysis been performed appropriately and rigorously? 

Reviewer #1: Yes

Reviewer #2: Yes

4. Have the authors made all data underlying the findings in their manuscript fully available?

Reviewer #1: Yes

Reviewer #2: Yes

5. Is the manuscript presented in an intelligible fashion and written in standard English?

Reviewer #1: Yes

Reviewer #2: Yes

6. Review Comments to the Author

Reviewer #1: A Confusion:

Its not clear that the Authors incorporated or not the below suggested References, and 2/3 files of the Article are merged, and it not also clear that which one is revised version, please.

6. Do add the following latest references in discussion part for justification of your results, please.

Zarei A (2017). Biochemical and pomological characterization of pomegranate accessions in fars province of Iran. SABRAO J. Breed. Genet. 49(2): 155–167.

Tarinta T, Chanthai S, Lertrat K, Nawata E, Techawongstien S (2020). Identification of the secondary metabolite capsiate in Capsicum germplasm accessions. SABRAO J. Breed. Genet. 52(2):144–157.

Upadyshev MT (2022). Apple cultivars and rootstocks assay for the identification of diverse viruses and healthy genotypes for breeding. SABRAO J. Breed. Genet. 54(1): 79-87. http://doi.org/10.54910/sabrao2022.54.1.8.

Motyleva SM, Medvedev SM, Morozova NG, Kulikov IM (2021). Leaf micromorphological and biochemical features of scab disease in immune and moderately resistant columnar apple (Malus domestica) cultivars. SABRAO J. Breed. Genet. 53(3): 352-366.

Hong-Qun, L. I., Li-Gang, X. I. N. G., & Xie-Ping, S. U. N. (2022). Predicting the potential distribution of Taxus wallichiana var. Mairei under climate change in China using maxent modeling. Pak. J. Bot. 54(4): 1305-1310.

Ahmad, N., Shakil, A., Shinwari, Z. K., Ahmad, I., & Wahab, A. (2022). Phytochemical study and antimicrobial activities of extracts and its derived fractions obtained from Berberis vulgaris L. and Stellaria media L. leaves. Pak. J. Bot. 54(4): 1517-1521.

Reviewer #2: (No Response)

7. PLOS authors have the option to publish the peer review history of their article (what does this mean?). If published, this will include your full peer review and any attached files.

Reviewer #1: **Yes: **PROF. DR. NAQIB Ullah KHAN

Reviewer #2: **Yes: **Reza Talebi

---

## [Author Response · Author response to Decision Letter 1]

20 Dec 2022

Dear Dr. Mehdi Rahimi, 

Dear Dr. Mehdi Rahimi, 

Thank you for giving me the opportunity to submit a revised draft of my manuscript titled "First assessment of Iranian pomegranate germplasm using targeted metabolites and morphological traits to develop the core collection and modeling of the current and future spatial distribution under climate change conditions”. I appreciate the time and effort that you and the reviewers have dedicated to providing your valuable feedback on our manuscript. I hope the manuscript after careful revisions meet PLOS ONE’s publication criteria.

I appreciate you for considering the revised version of the manuscript and look forward to receiving your decision.

Sincerely,

Mehrshad Zeinalabedini

Mehrshad Zeinalabedini, PhD

Agricultural Biotechnology Research Institute of Iran (www.abrii.ac.ir)

Mahdasht Road, Karaj, Iran. P.O. Box: 31535-1897 

Tel: +98 (261) 2703536 

Fax: +98 (261) 2704539 

Email: mzeinolabedini@abrii.ac.ir

---

## [Decision Letter · Decision Letter 2]

26 Dec 2022

PONE-D-22-06916R2First assessment of Iranian pomegranate germplasm using targeted metabolites and morphological traits to develop the core collection and modeling of the current and future spatial distribution under climate change conditionsPLOS ONE

Dear Dr. Zeinalabedini,

Thank you for submitting your manuscript to PLOS ONE. After careful consideration, we feel that it has merit but does not fully meet PLOS ONE’s publication criteria as it currently stands. Therefore, we invite you to submit a revised version of the manuscript that addresses the points raised during the review process.

We look forward to receiving your revised manuscript.

Kind regards,

Mehdi Rahimi, Ph.D.

Academic Editor

PLOS ONE

Journal Requirements:

Reviewers' comments:

Reviewer's Responses to Questions

**Comments to the Author**

1. If the authors have adequately addressed your comments raised in a previous round of review and you feel that this manuscript is now acceptable for publication, you may indicate that here to bypass the “Comments to the Author” section, enter your conflict of interest statement in the “Confidential to Editor” section, and submit your "Accept" recommendation.

Reviewer #1: (No Response)

2. Is the manuscript technically sound, and do the data support the conclusions?

Reviewer #1: (No Response)

3. Has the statistical analysis been performed appropriately and rigorously? 

Reviewer #1: (No Response)

4. Have the authors made all data underlying the findings in their manuscript fully available?

Reviewer #1: (No Response)

5. Is the manuscript presented in an intelligible fashion and written in standard English?

Reviewer #1: (No Response)

6. Review Comments to the Author

Reviewer #1: (No Response)

7. PLOS authors have the option to publish the peer review history of their article (what does this mean?). If published, this will include your full peer review and any attached files.

Reviewer #1: **Yes: **PROF. DR. NAQIB ULLAH KHAN

---

## [Author Response · Author response to Decision Letter 2]

10 Jan 2023

Dear Richard Ibañez Dilla

Many thanks for your email informing us to revise our manuscript.

We addressed the following issues. 

1. If possible, please upload an updated file showing your changes either highlighted or using track changes. This should be uploaded as a Revised Manuscript w/tracked changes, file type. Please follow this link for more information: http://blogs.PLOS.org/everyone/2011/05/10/how-to-submit-your-revised-manuscript/

Re. In response to this issue, it is necessary to explain that in the file "Response to reviewers" that we sent the last time, we stated that since there was no change in the manuscript, we uploaded two files 'Response to Reviewers' and 'Manuscript'. 

We upload new file "revised manuscript with track changes".

2. Please amend the manuscript submission data (via Edit Submission) to include all the authors.

Re: we amended the manuscript submission data to include all the authors

3. Please amend your list of authors on the manuscript to ensure that each author is linked to an affiliation. 

Re: we amended our list of authors on the manuscript so that each author is linked to an affiliation.

We hope that we have satisfactory tackled all issues raised and that the manuscript is now well suited for publication.

We appreciate you for considering the revised version of the manuscript and look forward to receiving your decision.

Sincerely,

Mehrshad Zeinalabedini

---

## [Editor Report · Decision Letter 3]

11 Jan 2023

First assessment of Iranian pomegranate germplasm using targeted metabolites and morphological traits to develop the core collection and modeling of the current and future spatial distribution under climate change conditions

PONE-D-22-06916R3

Dear Dr. Zeinalabedini,

We’re pleased to inform you that your manuscript has been judged scientifically suitable for publication and will be formally accepted for publication once it meets all outstanding technical requirements.

Kind regards,

Mehdi Rahimi, Ph.D.

Academic Editor

PLOS ONE
---

## [Editor Report · Acceptance letter]

16 Jan 2023

PONE-D-22-06916R3 

First assessment of Iranian pomegranate germplasm using targeted metabolites and morphological traits to develop the core collection and modeling of the current and future spatial distribution under climate change conditions 

Dear Dr. Zeinalabedini:

I'm pleased to inform you that your manuscript has been deemed suitable for publication in PLOS ONE. Congratulations! Your manuscript is now with our production department. 

Kind regards, 

on behalf of

Associate Prof. Mehdi Rahimi 

Academic Editor

PLOS ONE